# Uncertainty-Guided Optimization on Large Language Model Search Trees

## Abstract

Tree search algorithms such as greedy and beam search are the standard when it comes to finding sequences of maximum likelihood in the decoding processes of large language models (LLMs). However, they are myopic since they do not take the complete root-to-leaf path into account. Moreover, they are agnostic to prior knowledge available about the process: For example, it does not consider that the objective being maximized is a probability and thereby has specific properties like being bound in the unit interval. Taking a probabilistic approach, we define prior beliefs over LLMs' transition probabilities and obtain posterior beliefs over the most promising paths in each iteration. These beliefs are useful for defining a sample-based, non-myopic acquisition function that allows for a more data-efficient exploration scheme than standard search algorithms on LLMs. Crucially, unlike expensive simulation-based non-myopic methods like the Monte Carlo tree search, our method only requires samples from the beliefs. We discuss how to select the prior and the acquisition function, and demonstrate in experiments with various LLMs that our method achieves higher efficiency than recent baselines: Our method achieves the same or a higher likelihood while expanding fewer nodes.

## 1 Introduction

The decoding process of an autoregressive large language model (LLM) can be seen as finding an optimal path in a search tree. The number of such paths is exponential, often exceeding the computational budget required to examine them all. This inevitably leads to *computational uncertainty* (Hennig et al., 2022): an uncertainty that *could be* fully resolved if enough compute was available to examine all paths, but in practice is present due to the limited resources. Standard algorithms for LLM decoding, such as beam search (Koehn et al., 2003), completely ignore this uncertainty.

We posit that quantifying this uncertainty can be beneficial to ensure better explorations on LLMs' search trees. To this end, we incorporate computational uncertainty into the search process to guide it in a non-myopic fashion (i.e., accounting for our belief about the values of future nodes) and importantly, in a more data-efficient manner, akin to Bayesian optimization methods (BO, Garnett, 2023; Kushner, 1964; Močkus, 1975). BO methods are recognized for their data efficiency, not merely because they quantify uncertainty, but because they exploit the structural characteristics within that uncertainty. E.g., in continuous optimization problems, prior knowledge, such as the smoothness of a function, is often available through Gaussian processes (Rasmussen & Williams, 2005) or (Bayesian) neural networks (Hernández-Lobato et al., 2017; Kristiadi et al., 2023; 2024).

Unlike standard BO, however, LLM decoding is a structured, discrete optimization problem—its search space is the set of all root-leaf paths in a tree. Moreover, commonly, the values or rewards associated with each node are probabilities, bounded between $0$ and $1$. Hence, uncertainty quantification here is not as well-studied as in BO. We will, therefore, assume that rewards at a node of the search tree are components of a discrete distribution. The characteristic property we aim to exploit is the *concentration strength*: whether the Categorical distributions are all highly concentrated at a few realistic options, or if some of them are nearly uniform, making any individual option drastically less likely to be the optimal one. Intuitively, one would expect this to have a strong influence on the number of paths that need to be considered. For instance, when the distribution is concentrated, it is less likely under our belief that other paths will overtake later on and one can be more greedy.

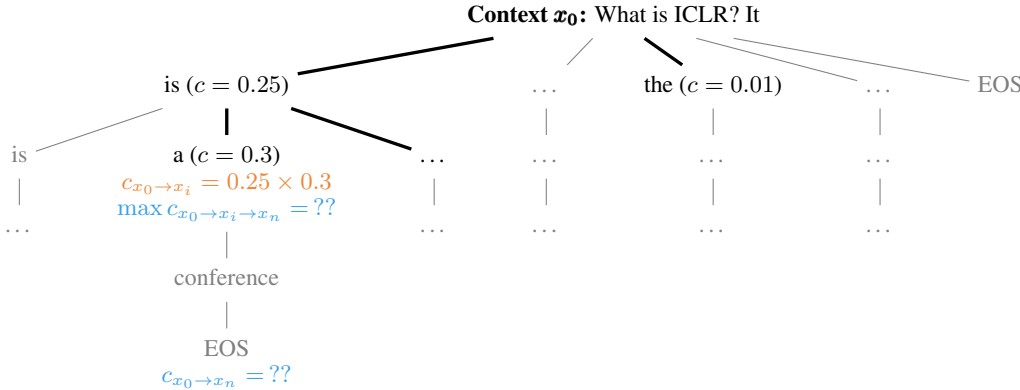

**Figure 1:** The intuition of our method—faded color represents unexplored subtrees. We sequentially expand a node of the LLM search tree based on the posterior belief. The latter is obtained by conditioning the unknown (the optimal total likelihood $\max c_{x_0 \to x_i \to x_n}$ from the root $\boldsymbol{x}_0$ to a leaf $x_n$ when continuing down a particular node $x_i$) on the current observations (the total likelihood $c_{\boldsymbol{x}_0 \to x_i}$ up until that node). This belief is induced by a simple prior belief on the LLM softmax probabilities $\boldsymbol{c}$. The samples of the posterior belief are used to decide which child node should be expanded next.

Meanwhile, when the distribution has high entropy, the uncertainty of our belief about "which next token is best" is higher and thereby requires more exploration and computational budget.

In this work, we propose a probabilistic framework that captures this aspect of the search space, which can help to decide which paths should be pursued and which can be ignored based on the posterior belief (Fig. 1). We do so by using a non-myopic acquisition function based on the samples of such a belief, which can be seen as a generalization of Thompson sampling (Thompson, 1933). Crucially, these samples are cheap, unlike other sample-based tree-search methods like Monte Carlo tree search. Moreover, we show that our acquisition function can easily be extended, e.g., to prefer sequences with higher diversity and less text degeneration (Holtzman et al., 2019).

Experiments on real-world text-generation benchmarks with various LLMs suggest that our method, called ***Uncertainty-guided Likelihood-Tree Search (ULTS)***, finds sentences with higher rewards than recent baselines with significantly fewer expensive node expansions. Moreover, ULTS only adds a small runtime overhead relative to the forward passes of the LLM. In summary:

 (i) We propose ULTS: a probabilistic decision-making framework on LLM search trees, where a prior belief is put on LLM softmax outputs. We show how to easily sample from the implied posterior over optimal values, and how to use these samples to make decisions when searching the tree.

 (ii) We demonstrate the efficiency and extensibility of ULTS in decoding recent LLMs.

 (iii) We open-source an implementation compatible with `transformers` (Wolf et al., 2020)

## 2 BACKGROUND

### 2.1 DECODING LARGE LANGUAGE MODELS

Let $\mathcal{A} = \{a_1, \ldots, a_b\}$ be a *vocabulary*—a set of natural-language tokens. A *large language model (LLM)* is a neural network that utilizes the attention mechanism (Vaswani et al., 2017), taking a *context*, an ordered sequence of tokens $\boldsymbol{x}_0 = (x_{01}, \ldots, x_{0k})$—each component takes values in the set $\mathcal{A}$—and outputs a distribution over $\mathcal{A}$, i.e. $\boldsymbol{c} = (c_1, \ldots, c_b)$ where each $c_i \geq 0$ and $\sum_{i=1}^{b} c_i = 1$.

The *de facto* way of generating a sequence of tokens given a context via an LLM $f$ is by generating each $x_{i+1}$ autoregressively given the previous sequence $(\boldsymbol{x}_0, x_1, \ldots, x_i)$. That is, for each $i = 1, \ldots, d$, we pick an $x_{i+1} \in \mathcal{A}$ according to the Categorical distribution $\boldsymbol{c}_{i(i+1)} := f(\boldsymbol{x}_0, x_1, \ldots, x_i)$ over the next token predicted by the LLM $f$ given the input $(\boldsymbol{x}_0, x_1, \ldots, x_i)$. We call this sequence-generating process a *decoding* process.

LLM decoding can be seen as a tree-search problem (Fig. 1). The context $\boldsymbol{x}_0$ is the root of the tree and at each step $i = 1, \ldots, d$, we are given a choice of $b$ many tokens to pick. This process is done recursively until termination; either when a specified depth $d$ has been reached or when a special token like "<EOS>" is selected. This means, the LLM search tree is an exponentially large tree (w.r.t. the number of tokens $b$ in the vocabulary) with depth $d$—the number of paths $\boldsymbol{x}_0 \to x_d$ from the root to leaves is $b^d$. Considering the fact that $b$ ranges from around 32k to 256k (Chowdhery et al., 2023; Radford et al., 2019), generating a sequence of tokens requires search over an intractably large space.

To address this problem, one can use a cheaper but heuristic way to explore this tree. The simplest way to do so is by sampling a token from the distribution $\boldsymbol{c}$ at each level of the tree (Holtzman et al., 2019). In some domains like machine translation and summarization, however, one often wants to find the sequence that maximizes the *total/joint* likelihood [1] $c_{\boldsymbol{x}_0 \to x_d} := \prod_{i=1}^{d} c_{(i-1)i}$ associated with each possible sequence $(\boldsymbol{x}_0, x_1, \ldots, x_d)$ (Wiher et al., 2022). This is an optimization-on-tree problem—heuristic optimization algorithms like greedy and beam search along with their variants (Freitag & Al-Onaizan, 2017; Meister et al., 2021; Vijayakumar et al., 2018, etc.) are the standard.

One can also view the decoding process as follows. On each of the tree's node $x_i$, we can define an optimal value $v_{x_i}$ associated with it: If $x_d$ is a leaf node, then $v_{x_d}$ corresponds to the total likelihood $c_{\boldsymbol{x}_0 \to x_d}$ from the root until that leaf node. Meanwhile, for inner nodes $x_i$ it is defined recursively in a bottom-up fashion as the maximum of the children's optimal values $\max_{x_c \in \text{children}(x_i)} v_{x_c}$. Intuitiveley, $v_{x_i}$ tells us "what is the total likelihood we would get if we continue descending the tree through the node $x_i$ in an *optimal* manner". The optimization-on-tree problem can then be recast as finding a path $\boldsymbol{x}_0 \to x_d$ that corresponds to the optimal value $v^*$ of the root. While for all $x_i$, the quantity $v_{x_i}$ cannot be feasibly computed, they are useful for our probabilistic treatment of LLM decoding in Section 3.

## 2.2 PROBABILISTIC DECISION-MAKING ON TREES

Decision-making under uncertainty requires a *belief* about the unknown given the current observation. This belief is then used to make decision, by computing an *acquisition function* which scores possible candidates. The candidate that has the highest acquisition value is then selected and the process is repeated until termination.

In the context of tree-structured problem, the approach to perform probabilistic decision-making is to perform probabilistic modeling over the unknown values $v_{x_i}$'s and use the posterior beliefs to make decision about which path should be followed next. In (Hennig et al., 2010), this kind of optimization has been done for game (e.g., Go) trees. They assume that the optimal value $v_{x_i}$ is decomposed into a latent score for the utility of the node if all remaining steps are taken randomly and additional increment to quantify how much better this score can get when all remaining steps are taken optimally instead of randomly. Their work relies on Gaussian process with a the Brownian motion kernel as their prior belief and has been extendes to more general directed-acyclic graphs in (Grosse et al., 2021).

# 3 ULTS: UNCERTAINTY-GUIDED LIKELIHOOD-TREE SEARCH

Here, we discuss *Uncertainty-guided Likelihood-Tree Search (ULTS)* which places a prior belief over any pre-trained LLM's softmax outputs and computes the implied posterior beliefs over the optimal likelihood values. We introduce and discuss the modeling assumptions in Section 3.1 and show how to derive approximate beliefs over the optimal values in the search tree in Section 3.2. The samples of the posterior beliefs are then used to make decisions about which subtree to expand next and when to stop the search (Section 3.4). Figure 3 gives an overview of our probabilistic model. Section 3 shows examples for the first two iterations of ULTS, including the observations, posterior beliefs and the acquisition functions.

---

[1]Implementation wise, this is usually done in the logarithmic space.

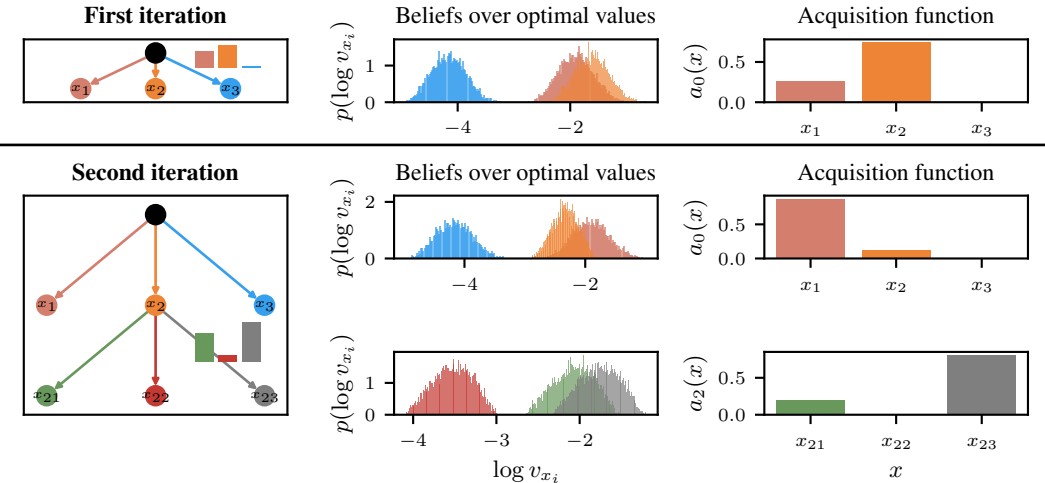

**Figure 2:** Two Example iterations with ULTS. The upper row show the observed categorical distirbution (left), the implied posterior over the optimal values in log space (center) and the resulting acquisition function over the children in the first level of the tree (right). The lower two rows show the corresponding quantities for the first and second level of the tree after the second iteration.

### 3.1 PRIOR BELIEFS OVER LLMS SOFTMAX OUTPUTS

For a node $x_i$ in an LLM search tree, let $\boldsymbol{c}_i = (c_{i1}, ..., c_{ib})$ be the vector containing the transition probabilities on the edges between $x_i$ and its $b$ children. Note that $\boldsymbol{c}_i$ defines a Categorical/discrete distribution, and we aim to exploit its structures by defining a prior belief over it.

A straightforward belief one can consider is the Dirichlet distribution. For tractability, we assume that the LLM's softmax probabilities are iid. draws from a symmetric Dirichlet distribution with parameter $\alpha > 0$, i.e., $p(\boldsymbol{c}_i) = \text{Dir}(\alpha)$. Thus, $\alpha$ controls how concentrated the sampled probability vectors are. In the context of LLMs, for small $\alpha$, the LLM would typically strongly favor a few tokens, whereas for large $\alpha$ the discrete distribution would closely resemble a uniform distribution over the tokens. The symmetry of the prior implies in our context that we do not have a preference for particular tokens *a priori*.

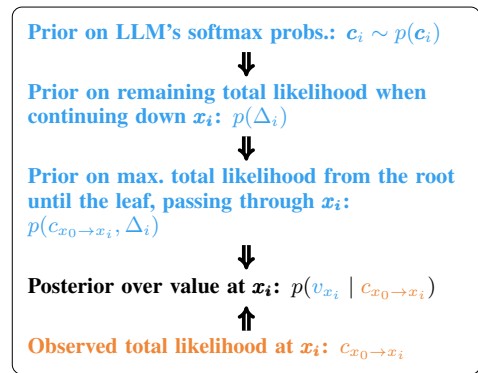

**Figure 3:** The beliefs we consider.

Another belief that we can consider is an empirical prior over $\boldsymbol{c}_i$. Let $\{\boldsymbol{x}_n\}_{n=1}^N$ be samples of contexts, e.g. a subset of the training/validation data. Given an LLM $f$, we can then obtain the set $\{(\boldsymbol{x}_i, x_{i1}, \ldots, x_{id})\}$ of $d$-step completions of $\boldsymbol{x}_i$'s through the LLM (e.g. through a greedy decoding). We can then collect samples of the Categorical distributions from this generation process: $\mathcal{C} = \coprod_{n=1}^N \{\boldsymbol{c}_{n1}, \ldots, \boldsymbol{c}_{nd}\}$. Then, instead of sampling from $\text{Dir}(\alpha)$, we can sample from $p(\boldsymbol{c}_i) = \text{Unif}(\mathcal{C})$. This prior is more flexible than the Dirichlet prior since no symmetric nor unimodality assumptions are made at the cost of more compute and memory overhead. Note, however, that all these priors are precomputed and can be reused across subsequent decoding runs, so they incur a fixed $\mathcal{O}(1)$ cost.

### 3.2 PRIOR BELIEFS OVER OPTIMAL VALUES

Having picked a prior over the LLM's softmax outputs, we can derive the implied priors over the optimal values $v_{x_i}$ for each node $x_i$ in the tree (Hennig et al., 2010). From the definition of an optimal value (Section 2), it factorizes as the product $c_{\boldsymbol{x}_0 \to x_i}$ of the transition probabilities from the root node

$x_0$ to $x_i$ and a remaining term which we refer to as $\Delta_i$, i.e., we have $v_{x_i} = c_{\boldsymbol{x}_0 \to x_i} \cdot \Delta_i$. Intuitively, the term $\Delta_i$, quantifies the likelihood that we get in the remaining steps from $x_i$ to a leaf node when we take all remaining decisions optimally. It can be defined by the following recurrence relation:

$$\Delta_i = \begin{cases} 1 & \text{if } x_i \text{ is a leaf,} \\ \max\limits_{x_j \in \text{children}(x_i)} \{c_{ij} \cdot \Delta_j\} & \text{otherwise.} \end{cases}$$

Due to the iid. assumption above (Section 3.1), we have the joint distribution $p(c_{\boldsymbol{x}_0 \to x_i}, \Delta_i) = p(c_{\boldsymbol{x}_0 \to x_i}) \, p(\Delta_i)$. Note that all these quantities are not analytically available. Thankfully, sampling from the *posterior* belief $p(v_{x_i} \mid c_{\boldsymbol{x}_0 \to x_i})$ is easy if we are able to sample from $p(\Delta_i)$ (Section 3.3).

Let us, therefore, derive an approximate sampling scheme for $p(\Delta_i)$ (Alg. 1). It follows Grosse et al. (2021); Hennig et al. (2010) who used Gaussian priors. We recursively approximate the prior distribution of the $\Delta_i$'s at level $l$ with Beta distributions $\mathcal{B}_l(\Delta_i)$. In a bottom-up approach, we generate samples $\{\max_j c_{nj} \mid c_n \sim p(\boldsymbol{c})\}_{n=1}^{N}$ for a $\Delta_i$ at level $l = d - 1$. Using these samples, we empirically fit the parameters of the Beta distribution $\mathcal{B}_{d-1}(\Delta_i)$ via maximum likelihood (AbouRizk et al., 1994). The distributions of the $\Delta_i$ are the same for all nodes on the same level due to the iid. assumption, so this has to be done only once. Note that we need this approximation since the distribution of the maximum $\max_j c_{ij}$ has no known analytic solution. We then continue by recursively sampling sets of the form (one per level) $\{\max_j c_{nj} \cdot \Delta_j \mid c_n \sim p(\boldsymbol{c}), \Delta_j \sim \mathcal{B}_{l+1}(\Delta)\}_{n=1}^{N}$ for a $\Delta_i$ of the level $l$ and using it to fit the parameters of $\mathcal{B}_l(\Delta_i)$. The time complexity is $\mathcal{O}(d \cdot b \cdot N)$ for computing the approximations, i.e., it is linear in the depth and width of the tree. We emphasize that they can be pre-computed before the search and reused across different decoding runs. Alg. 1 in Appendix D contains pseudoscode for this sampling scheme.

### 3.3 Posterior beliefs over optimal values of frontier nodes

Notice that whenever a new node $x_i$ is added to the search tree, the likelihood associated with the path from the root to that node $c_{\boldsymbol{x}_0 \to x_i}$ is fully observed—its distribution is simply a Dirac delta. This means, the joint distribution becomes $p(c_{\boldsymbol{x}_0 \to x_i}, \Delta_i) = \delta(c_{\boldsymbol{x}_0 \to x_i}) \, p(\Delta_i)$. Therefore, to sample $v_{x_i}$ given that we have observed $c_{\boldsymbol{x}_0 \to x_i}$—i.e., sampling from the posterior $p(v_{x_i} \mid c_{\boldsymbol{x}_0 \to x_i})$—it is sufficient to sample from $p(\Delta_i)$ and then simply scale all samples by $c_{\boldsymbol{x}_0 \to x_i}$. During the search, these samples are backed up the tree and leveraged to make decisions in exploring the tree. Notice that these samples are cheap to get, unlike empirical samples based on simulations/rollouts.

### 3.4 Decision making

At each iteration, we use the posterior samples by following the steps of Monte Carlo tree search, but without the expensive rollout step. Alg. 2 in Appendix D contains the pseudocode for the decoding portion of ULTS.

**1. Selection** Starting from the root node $\boldsymbol{x}_0$, we recursively pick a child based on an acquisition function until an unexpanded node is selected. From a decision-theoretic perspective, this is done by choosing a utility function that encodes our preferences about the outcome (e.g. high likelihood) and then integrating out the unknown variables influencing this outcome. Let $\boldsymbol{v}_i$ contain the (unknown) optimal values of the descendants of $x_i$ and let $u(x_c, \boldsymbol{v}_i)$ be a utility function. The idea is to select the child node $x_c$ that maximizes the expected utility:

$$\int u(x_c, \boldsymbol{v}_i) \, p(\boldsymbol{v}_i \mid c_{v_{\boldsymbol{x}_0 \to x_i}}) \, d\boldsymbol{v}_i, \tag{1}$$

For ULTS, we use an utility function that encodes our preference for finding a descendant with high optimal value: $u(x_c, \boldsymbol{v}_i) := \mathbb{I}[v_{\hat{x}_c} = \max_{x_j \in \text{children}(x_i)} v_{\hat{x}_j}]$. Different realizations of $\hat{x}_c$ and $\hat{x}_j$ can be used: The most straightforward is to use the children's optimal values themselves. This corresponds to simply setting $\hat{x}_j = x_j$ and $\hat{x}_c = x_c$. Another strategy is to use the optimal values of each child's best descendant—intuitively, it performs a "lookahead". In this case, $\hat{x}_j$ is defined as (similarly for $\hat{x}_c$):

$$\hat{x}_j = \begin{cases} x_j & \text{if } x_j \text{ is a leaf,} \\ \arg\max\limits_{x_c \in \text{children}(x_j)} a_j(x_c) & \text{otherwise.} \end{cases} \tag{2}$$

Note that they have the same costs since beliefs over the optimal values for both strategies are readily available due to the backup step below, i.e., we do not actually perform the recursion (2) in this step. The acquisition function is then derived as a sample-based approximation to the expectation in Eq. 1:

$$a_i(x_c) = \frac{1}{N} \sum_{n=1}^{N} \mathbb{I}\big[(v_{\hat{x}_c})_n = \max_{x_j \in \text{children}(x_i)} (v_{\hat{x}_j})_n\big]. \tag{3}$$

We can replace the above utility function with *any* other utility function and then use the posterior samples to derive the acquisition function (Wilson et al., 2018). For example, we can use a utility function that encodes our preference of picking sequences with minimal repetition, which is desirable to avoid text degeneration (Holtzman et al., 2019). Concretely, we can employ a new utility function that contains a repetition-penalty term:

$$\tilde{u}(x_c, \boldsymbol{v}_i) = \mathbb{I}[(\log v_{\hat{x}_c} + \lambda\, b(\hat{x}_c)) = \max_{x_j \in \text{children}(x_i)} (\log v_{\hat{x}_j} + \lambda\, b(\hat{x}_j))], \tag{4}$$

where $b(\hat{x}_c)$ is a log-diversity term (Su et al., 2022).

**2. Expansion** Given an unexpanded node $x_i$, we query the LLM to obtain the top-$k$ most likely children and their corresponding likelihoods. These likelihoods are new observations and we combine them with the prior samples to obtain $n$ posterior samples $(v_{x_c})_n$ of each child $x_c$. This step is the most expensive part of any tree-search algorithm since querying the LLM is costly. Akin to BO, ULTS strives to reduce the number of node expansions and thus minimize the number of LLM calls.

**3. Backup** We recursively propagate the newly obtained posterior samples $\{(v_{x_c})_n\}_c$ back up the tree until the root via the path selected in the previous steps. This is done to update the posterior samples contained in each node of the path. They will then influence the selection process in the next iteration, updating the exploration-exploitation tradeoff. Different update strategies can be used depending on the choice of the acquisition function (3). When posterior samples of the children node $x_c$ are used to compute $a_i(x_c)$, then we propagate up the posterior samples of the newly expanded nodes as in when computing the prior by recursively taking their maximum, i.e. the $i$-th sample for the optimal value of parent node $x_p$ is given by the maximum over the $i$-th sample of the children's optimal values:

$$(v_{x_p})_i = \max_{x_c \in children(x_p)} (v_{x_c})_i. \tag{5}$$

When the posterior samples of the best descendant of $x_c$ are used in $a_i(x_c)$, we simply propagate up the posterior samples of the best child among the newly expanded nodes, without taking further maximums along the path:

$$(v_{x_p})_i = (v_{x_c*})_i, \qquad \text{where } x_c^* = \arg\max_{x_c \in \text{children}(x_p)} a_p(x_c). \tag{6}$$

**Termination criterion** Finally, the posterior samples over the optimal values can not only be used for the selection of new nodes but also to monitor the progress of the optimization. For instance, one can compute the following empirical probability $\hat{\mathbb{P}}(c^* < v_{\boldsymbol{x}_0}) = \frac{1}{N} \sum_{n=1}^{N} \mathbb{I}[c^* \leq (v_{\boldsymbol{x}_0})_n]$, which corresponds to the probability of the current best likelihood $c^*$ among all leaves ULTS has visited so far is lower than the best value $v_{\boldsymbol{x}_0}$ at the root, according to our posterior belief. Note that the computation of such a probability is done as in the acquisition function $a(x)$, i.e., using the posterior samples of the root node $\boldsymbol{x}_0$ or the posterior samples of the best descendant. Then, one can decide to stop the search once this probability is below some confidence level $\varepsilon > 0$.

### 3.5 REMARKS

**Practical considerations** In order to put a tractable upper bound on the runtime of ULTS, we introduce a hyperparameter $k_{\max}$ on the maximum number of nodes that can be expanded per level, similar to beam search or top-$k$ sampling. Moreover, we stop the search as soon as a set of $k_{\max}$ leaves is attained or the termination probability exceeds below $1 - \varepsilon$. We shall see in Section 5 that even under these further constraints, we still obtain good results while being efficient.

**Limitations** There are strong reasons to assume that the iid. assumption might not fully hold. Indeed, our modeling assumptions are chosen with practicality in mind. This is akin to how priors

in Bayesian neural networks (Wilson & Izmailov, 2020) are chosen, which are usually simply just isotropic Gaussians. Even in standard BO, a generic Gaussian-process prior/kernel is often assumed by default (Balandat et al., 2020). For a further discussion of the prior assumptions, see Appendix A. Moreover, our work does not take the miscalibration of LLM softmax outputs into account. Indeed, it has been shown that higher likelihood is not always correlated with human preferences (Eikema & Aziz, 2020; Holtzman et al., 2019; Stahlberg & Byrne, 2019; Wiher et al., 2022; Zhang et al., 2020). However, LLM calibration is orthogonal to the focus of the present paper—we leave this for future work. Recent work from (Yoshida et al., 2023), for example, proposes to use a likelihood conditioned on certain attributes of the sequence and to which ULTS based decoding would directly be applicable. Nevertheless, ULTS is extensible: One can construct a different prior or use a different utility function to address this; see Appendix C.8 for example.

## 4    RELATED WORK

Commonly employed stochastic decoding strategies are nucleus sampling (Holtzman et al., 2019) and best-of-k sampling (Stiennon et al., 2020). Their purpose is to make the decoding processes more robust towards potential degeneration in the likelihood learned by the LLM (also see Section 3.5). I.e, these strategies address uncertainty about the objective function, whereas we focus on epistemic uncertainty due to limited compute resources. Another decoding method similar to ULTS is Minimum Bayes Risk (MBR) decoding (Eikema & Aziz, 2021; Kumar & Byrne, 2004). This strategy also relies on expected maximization of a utility function (typically not one that results in likelihood maximization, even though possible), but it does not employ a probabilistic model to estimate the unknown, but rather samples a set of hypothesis sequences from the LLM directly, which is more expensive. We point out that ULTS can be combined with MBR by using ULTS to generate a set of hypothesis sequences for MBR.

Probabilistic/Bayesian optimization on trees has been proposed for game trees. Hennig et al. (2010) developed a roll-out based probabilistic tree search algorithm for game trees, e.g. for playing Tic-Tac-Toe and Go. Crucially, the structure of the game tree is different than the LLM search tree. Moreover, they assumed a Gaussian process prior with the Brownian motion kernel in conjunction with the expectation propagation algorithm (Minka, 2001) to model their beliefs. Grosse et al. (2021) extended their approach for a more general directed acyclic graph structure. In contrast, we focus on the tree implied by the sequential generation process in LLMs with a Dirichlet or empirical prior.

Our method can be seen as utilizing a best-first search strategy, of which the A* algorithm (Hart et al., 1968) is the most famous. Moreover, an A*-like beam search algorithm, under the name of best-first beam search, has also been proposed for decoding LLMs (Meister et al., 2020). Unlike the non-probabilistic A* and best-first beam search, we approach optimization-on-tree problems via the lens of decision-making under uncertainty—putting a prior belief about the unknown, updating it based on the observations, and making decision based on the posterior belief. Furthermore, best-first beam search (Meister et al., 2020) has different goals to ours: it is designed to output the same set of leaves (and thus the optimal likelihood value) as the standard beam search. ULTS, meanwhile, focuses on both performance (i.e., attaining higher likelihood than beam search) and efficiency (reducing the number of costly LLM forward passes).

Finally, Monte Carlo tree search, which also utilizes samples to make decisions, has also been proposed for decoding LLMs (DeLorenzo et al., 2024; Feng et al., 2023; Hao et al., 2023; Leblond et al., 2021; Liu et al., 2023; Zhang et al., 2023; Zhou et al., 2024). Different from our goal, they focused on incorporating external rewards that are only observable at the leaf nodes. For instance, Zhang et al. (2023) defined the reward to be the unit-test results for code-generating LLMs. Moreover, Monte-Carlo tree search does not define a probabilistic model (i.e. a prior and posterior beliefs) over quantities in the search tree—its decision making is based on the statistics obtained by exploring the search tree on the fly. In contrast, ULTS does not require costly gathering of those statistics during the decoding process; instead, it makes decisions based on precomputed samples from its beliefs.

## 5    EXPERIMENTS

**Setup**    In this section, we evaluate ULTS in both close-ended and open-ended text generation problems. We set ULTS' $k_{\max} \in \{2, 3, 4, 5, 10, 20\}$ and set $\epsilon$ to a default value of $0.1$. The

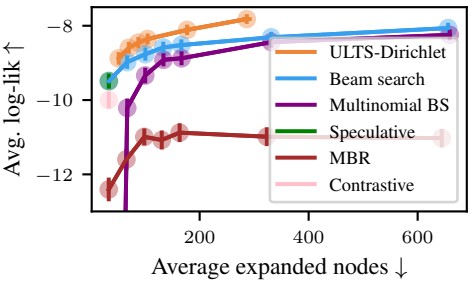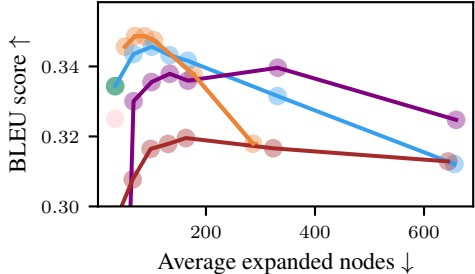

**Figure 4: Left:** Machine translation results with the WMT-19 English-to-German dataset in term of log-likelihood. **Right:** The corresponding BLEU scores. For MBR, the average number of expanded nodes is only an estimate based on the number of generated hypothesis (token sequences) times the length of the final selected hypotheses.

error bars indicate $\pm 1$ standard error of the mean based on all test sentences. We compare ULTS against beam search (the *de-facto* decoding method for machine translation), as well as the recent baselines multinomial beam search (Multinomial BS; Kool et al., 2019), contrastive search (Su et al., 2022) nucleus sampling (Holtzman et al., 2019), best-of-k sampling (Stiennon et al., 2020), speculative decoding (Leviathan et al., 2023), and DoLA (Chuang et al., 2024). For beam search, multinomial beam search, and best-of-k baselines, we evaluate beam sizes and numbers of samples $k \in \{1, 2, 3, 4, 5, 10, 20\}$, respectively. We set any other hyperparameters of all baselines as suggested by the original papers or by Huggingface. We use Huggingface's implementation of the baselines and provide our implementation of ULTS in the supplements.

**Choice of the prior** We sampled a set $\mathcal{C}$ of Categorical distributions from the LLM on training sequences for each of the datasets and both of the LLMs. We used 1000 samples from the training set for each of the datasets and LLMs we considered. They are used for the empirical prior, as well as training data for fitting the concentration parameter $\alpha$ of the Dirichlet prior. Figure 8 in Appendix C.3 shows samples for the maximum of Categorical distributions returned by the LLM, as well as the distribution of the maximum of the Categorical distribution from a Dirichlet prior for $\alpha = 10^{-1}, 10^{-4}, 5 \times 10^{-6}$. For the experiments below, we picked a value for $\alpha$ based on the histograms in Figure 8. For a comparison between different values of the concentration parameter, please refer to Fig. 9 (Appendix C.3). Note that this way of choosing the hyperparameter $\alpha$ is very convenient compared to performing costly cross-validation.

## 5.1 CLOSE-ENDED GENERATION

We compare ULTS with the baselines in a machine translation task with 1000 randomly sampled sequences from the WMT-19 English to German dataset in Section 5 in terms of likelihood and BLEU score. For the LLM, we use T5-large (Raffel et al., 2020); for ULTS, we use a Dirichlet prior with $\alpha = 5 \times 10^6$. Unless specified otherwise, we use the strategy in (2) for the selection and backup steps. See Appendix C for results with the other strategy and further dataset/setting. All experiments are done on a single NVIDIA GeForce RTX 2080 Ti and NVIDIA A40 48GB GPUs for GPT-2 and Llama-2-7b, respectively. We stop the exploration of a path in the tree if the <EOS> token is found or when the maximum depth of 60 is exceeded. The results are in Figure 3. ULTS is both more efficient (fewer node expansion) and more performant (higher average log-likelihood) at all values of $k/k_{\max}$. For all methods, small to medium beam sizes of $k = 3$ seem to work best in terms of BLEU scores—these scores decrease for a larger (maximum) beam with, in contrast to the log-likelihood in Section 5 . See Appendix C.2, for qualitative examples of the translated sentences.

In addition, we test ULTS for code completion using the LLM CodeLlama-7b-Python-hf (Lu et al., 2024) and all 164 sequences from the OpenAI HumanEval dataset (Chen et al., 2021). The maximum tree depth is set to 500 tokens. We use a Dirichlet Prior with $\alpha = 5e - 6$. In order to allow for a task-specific qualitative comparison, we evaluated the generated token sequences w.r.t. the Pass@1 metric (i.e. percentage of test cases passed), see Table 1. ULTS achieves similar performance in terms of Pass@1 while and expanding fewer nodes and being faster in wall-clock time.

| Method | Pass@1 (%) ↑ | Log-likelihood ↑ | Node expansions ↓ | runtime (s) ↓ |
|---|---|---|---|---|
| Greedy | 14.02 | -28 | 399.369 | $13.557 \pm 0.493$ |
| Nucleus sampling | 14.63 | -28.5 | 377.671 | $10.568 \pm 0.429$ |
| Beamsearch-Mult ($k = 2$) | 27.44 | -24.375 | 496.963 | $7.235 \pm 0.420$ |
| ULTS ($k_{max} = 2$) | 25.00 | -16.625 | 134.835 | $4.447 \pm 0.421$ |
| ULTS ($k_{max} = 3$) | 34.76 | -15.5 | 146.866 | $4.936 \pm 0.454$ |
| ULTS ($k_{max} = 4$) | 32.32 | -15.688 | 161.634 | $5.180 \pm 0.398$ |
| ULTS ($k_{max} = 5$) | 32.93 | -14.938 | 179.39 | $5.596 \pm 0.358$ |
| ULTS ($k_{max} = 10$) | 31.71 | -12.375 | 219.994 | $6.988 \pm 0.464$ |
| ULTS ($k_{max} = 20$) | 23.78 | -12.062 | 180.36 | $6.041 \pm 0.369$ |

**Table 1:** Results for the code completion task on the HumanEval dataset.

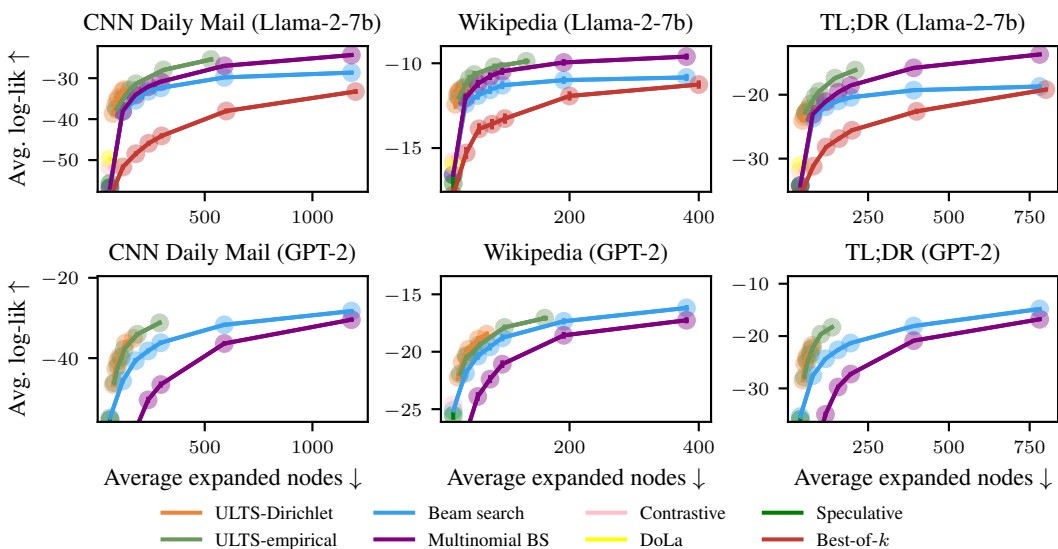

**Figure 5:** Decoding experiments with Llama-2-7b and GPT-2 for text generation on CNN Daily Mail and Wikipedia articles and text summarization for the TL;DR dataset. The methods are evaluated for different computational budgets, i.e. different values of $k$ and $k_{max}$. In our comparison, ULTS lies at the Pareto frontier of all methods we benchmark.

## 5.2 OPEN-ENDED GENERATION

We use GPT-2 (Radford et al., 2019) and Llama-2-7b (Touvron et al., 2023) for text generation on articles from the Wikipedia (See et al., 2017), CNN Daily Mail (Hermann et al., 2015), and Reddit TL;DR (Völske et al., 2017) datasets. Since many of the text samples in the Wikipedia dataset end with e.g., references instead of full sentences, we filter for text samples with at least 500 tokens, resulting in a test set with 332 token sequences (out of a random subset of originally 1000 sequences). We use 200 tokens as input and predict 20 tokens. We do the same for the CNN Daily Mail dataset, where we end up with 790 token sequences. We use a context length of 300 and generate 60 tokens. We also include a summarization task, where the goal is to generate a 40 token long summary of the input sequence. For this, we use 1000 random samples from the TL;DR dataset with variable-length contexts. Here, we decode sequences of fixed length instead of stopping at the <EOS> token. We run ULTS with both the Dirichlet prior with $\alpha = 10^{-4}$ and the empirical prior.

Figure 5 shows the results. Baselines underperforming in our setting are shown in Figure 12 in the Appendix. No matter the choice of $k_{max}$ (and $k$), ULTS yields sequences with the same or a higher log-likelihood while expanding fewer nodes. For example in the summarization task with GPT-2, ULTS with the empirical prior and $k_{max} = 20$ achieves an average log-likelihood of $-18.31$ while expanding only 137.90 nodes. In contrast, beam search with $k = 5$ returns sequences of average log-likelihood $-21.33$ even though expanding more nodes (196 nodes). This is the case for both

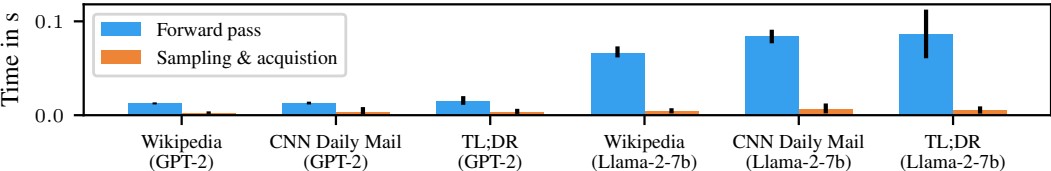

**Figure 6:** Average wall-clock time per iteration.

choices of priors, with the empirical prior encouraging exploration a bit more. The histogram in Fig. 7 (Appendix C.3) suggests that the distribution $\max_i c_{ji}$ is bimodal and not fully captured by the Dirichlet prior. In particular, our choice of Dirichlet prior is slightly too pessimistic. As a result, the search may stop too early and the available budget may not be fully utilized. However, the budget that *is* used, is used efficiently. Our recommendation is thus as follows. When efficiency is the main goal, a Dirichlet prior with low concentration is preferable—it performs similarly to beam search with smaller beam widths, while being more efficient. If the search performance is important and an additional tree exploration can be afforded (still more efficient than beam search), then the empirical prior is the best choice—it is also hyperparameter-free.

Additional results under the ROUGE metric can be found for the summarization task in Appendix Section C.9. We also tested ULTS with the utility function $\tilde{u}$ with an additional diversity penalty introduced in Eq. (4). Details and results can be found in Table 4 in Appendix C.8. ULTS can achieve good perplexity and diversity at the same time, showing the flexibility of our probabilistic decision-theoretic perspective.

### 5.3 RUNTIME

We analyze the runtime overhead on top of the LLM's forward pass due to ULTS. Figure 6 shows the wall-clock time per iteration, averaged over all sentences and all iterations, broken down into the time spent on the LLM's forward pass and the time spent on sampling the optimal values and optimizing the acquisition function. The error bars indicate the $95\%$- confidence intervals. Results are shown for the experiments with $k_{\max} = 20$ and the Dirichlet prior—the empirical prior performs similarly since ULTS does not differentiate between them in Alg. 2. We note that the runtime overhead of ULTS is small compared to the time spent to do an LLM forward pass. However, note that our current implementation only expands one node of the search tree in each iteration and thereby only evaluate one token sequence per forward-pass. ULTS can be extended similar like BO can be extended into batch BO. This is outside of the present work's scope but is a promising direction for future work.

### 6 CONCLUSION

We have discussed our method, ULTS, a probabilistic decision-making algorithm for the decoding process of large-language-model (LLM) search trees. Our method quantifies and leverages computational uncertainty over the maximum value of the optimization process. ULTS exploits the structure of the optimization problem—in particular, the concentration strength of the LLM softmax outputs—by putting a prior over the LLM's softmax outputs and using the implied posterior samples to guide exploration-exploitation in a non-myopic manner. Our work thus opens up opportunities for interesting future work in probabilistic inference for LLMs.

**Future work** One can study the effect of using more sophisticated priors, e.g. where iid. is not assumed. One can also consider batched acquisition strategies, similar to batch Bayesian optimization techniques (González et al., 2016; Wu & Frazier, 2016). Moreover, it is also interesting to incorporate the uncertainty over the LLM's outputs (e.g. in the context of Bayesian LLMs) in order to account for possible miscalibration of the likelihood. Another research direction is to extend the approach to reasoning tasks beyond language tasks, e.g. by including beliefs over external rewards (Zhang et al., 2023), such as ones coming from an RLHF-trained reward model, in addition to the current likelihood rewards. Finally, our work in connecting probabilistic inference with LLMs paves the way to perform probabilistic reasoning with the tree of thoughts framework (Yao et al., 2023).

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

## APPENDIX A  FURTHER DISCUSSIONS

### A.1  PRIOR ASSUMPTIONS

The symmetry assumption for the Dirichlet prior on the LLM's softmax outputs likely does not entirely hold—some words occur more often in natural language than others (Kingsley Zipf, 1932). However, ULTS only requires access to the distribution over the *maximum* of the unexplored part of the search space and not over the *arg max*. The former is not affected by permutations of the entries in the categorical distributions, which is why we suspect that a symmetric Dirichlet distribution with sufficiently small concentration parameter is a good proxy. Moreover, the empirical prior can also be used to address this limitation, without incurring large overhead.

We assume the same prior for all sentences in a dataset. It could be, though, that some input prompts are easier to complete than others, and the corresponding LLM's outputs are therefore generally have less entropy than on other those of other prompts. This could be counteracted by choosing a personalized hyper-parameter $\alpha$. This would require deriving the prior for multiple possible values of $\alpha$, which scales linearly with the number of possible values for $\alpha$. However, this can be precomputed (as in Alg. 1) such that it would not affect the costs during inference.

## APPENDIX B  ADDITIONAL EXPERIMENTAL DETAILS

URLs to the models and datasets used are provided below:

- Models:
    - https://huggingface.co/meta-llama/Llama-2-7b-hf
    - https://huggingface.co/openai-community/gpt2
    - https://huggingface.co/google-t5/t5-large
    - https://huggingface.co/codellama/CodeLlama-7b-Python-hf
- Datasets:
    - https://huggingface.co/datasets/wikipedia
    - https://huggingface.co/datasets/cnn_dailymail
    - https://huggingface.co/datasets/CarperAI/openai_summarize_tldr
    - https://huggingface.co/facebook/wmt19-de-en
    - https://huggingface.co/datasets/openai/openai_humaneval

### B.1  HYPERPARAMETERS

For beam search, multinomial beam search there are no hyperparameters beyond the beam size $k$. Speculative Search uses greedy search with n-gram based assisted decoding with prompt_lookup_num_tokens = 10. Best-of-k Sampling, we use top-p=0.95. For Nucleus Sampling, we use a temperature parameter of 0.2 and top-p=0.95. For contrastive search, we use penalty parameter $\alpha = 0.6$ and top-k=50. For DoLA, we set the number of DoLA layers to "high". We use version 4.38.2. of the transformers package. For MBR decoding, we use the implementation available at https://github.com/ZurichNLP/mbr (Bertsch et al., 2023; Vamvas & Sennrich, 2024) with the default utility "fastChrF", temperature parameter $0.5$ and number of sampled hypothesis is $\{1, 2, 3, 4, 5, 10, 20\}$.

## APPENDIX C  ADDITIONAL EXPERIMENTAL RESULTS

### C.1  ON-MODEL EXPERIMENTS

We compare ULTS to beam search on artificially generated search problems from Dirichlet priors. The trees have branching factor $b = 8$ and depth $d = 5$. Since these trees are so small we optimized

the acquisition function in eq. (3) over the the full boundary, i.e. non-recursively. The transition probabilities at each node are sampled from a Dirichlet prior with fixed $\alpha \in \{0.1, 0.2, 0.5, 0.8\}$. The comparison is on-model, i.e., ULTS is run with the ground truth parameter of $\alpha$. We repeat the experiment with different values for the confidence parameter $\varepsilon$ of ULTS from $\{0.05, 0.1, 0.3\}$. Since the toy problems are so small, the exploration of too many nodes is not an issue and we use $k_{\max} = \infty$. Beam search is run with beam sizes ranging from 1 to 7. The results in Fig. 7 show that ULTS dominates across the entire range of hyper-parameters. This suggests that knowledge about concentration strength helps reduce the number of search steps.

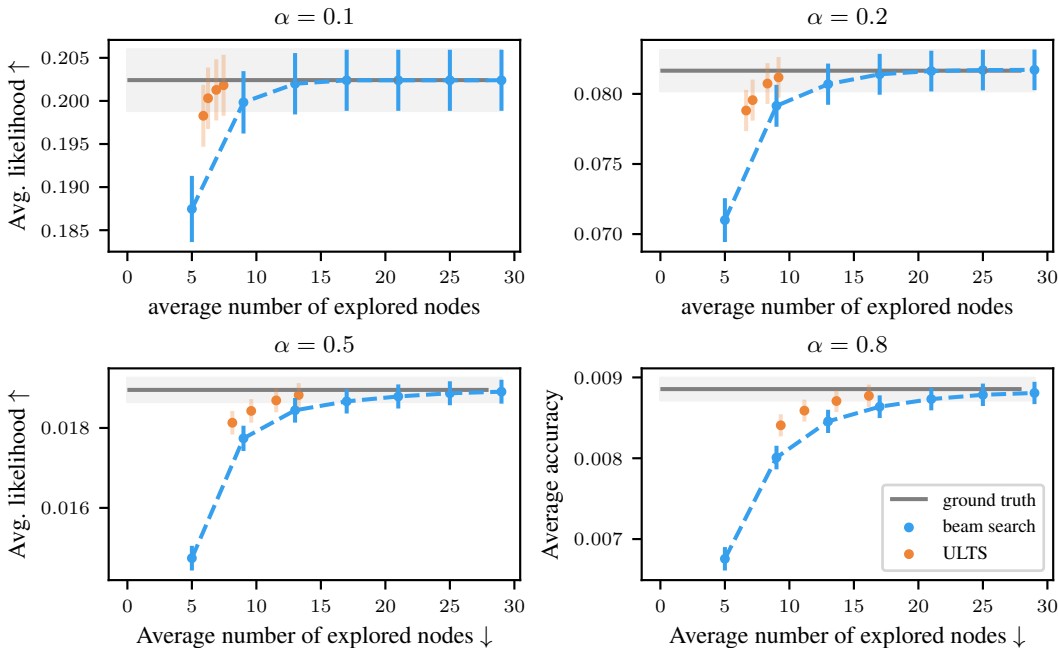

**Figure 7:** Comparison on trees, where the transition probabilities are sampled from a Dirichlet prior for different values of the concentration parameter.

## C.2 MACHINE TRANSLATION

We show some example sequences found with ULTS and beam search. For these examples, we picked sentences where ULTS achieves better BLEU scores. Other examples have mainly the same outputs as beam search while ULTS is more efficient.

---

**Example 1**

**input prompt:** *It is annoying when geographical maps are not up-to-date.*
**ground truth:** *Es nervt, wenn Landkarten nicht aktuell sind.*

**ULTS translation:** *Es ist ärgerlich, wenn geographische Karten nicht aktuell sind.*
**ULTS BLEU/log-likelihood:** 0.189/-3.914

**Beam search translation:** *Es ist ärgerlich, wenn die geographischen Karten nicht auf dem neuesten Stand sind.*
**Beam search BLEU/log-likelihood:** 0.0/-4.230

---

---

**Example 2**

**input prompt:** *The historical maps of the digital BayernAtlas, an offering from the State Government's Geoportal Bayern, are anything but up-to-date – and yet it is precisely for this reason that they are so informative.*
**ground truth:** *Die historischen Landkarten des digitalen Bayern-Atlases, ein Angebot des Geoportals Bayern der Staatsregierung, sind alles andere als aktuell - doch gerade deshalb sehr aufschlussreich.*

**ULTS translation:** *Die historischen Karten des digitalen BayernAtlas, ein Angebot des Landesgeoportals Bayern, sind alles andere als aktuell – und gerade deshalb so informativ.*
**ULTS BLEU/log-likelihood:** 0.292/-9.247

**Beam search translation:** *Die historischen Karten des digitalen BayernAtlas, ein Angebot des Landesgeoportals Bayern, sind alles andere als aktuell – und gerade deshalb sind sie so informativ.*
**Beam search BLEU/log-likelihood:** 0.272/-9.59

---

**Example 3**

**input prompt:** *Even if the French troops finally retreated with the Treaty of Lunéville from 9th February 1801: it was the current neighbours who had the idea to create a comprehensive map of Bavaria.*
**ground truth:** *Auch wenn die französischen Truppen mit dem Frieden von Lunéville vom 9. Februar 1801 schließlich abzogen: Es waren die heutigen Nachbarn, die die Idee einer flächendeckenden Bayern-Karte kreierten.*

**ULTS translation:** *Auch wenn sich die französischen Truppen mit dem Vertrag von Lunéville vom 9. Februar 1801 schließlich zurückzogen: Es waren die heutigen Nachbarn, die die Idee hatten, eine umfassende Karte von Bayern zu erstellen.*
**ULTS BLEU/log-likelihood:** 0.458/-13.111

**Beam search translation:** *Auch wenn sich die französischen Truppen schließlich mit dem Vertrag von Lunéville vom 9. Februar 1801 zurückzogen: Es waren die heutigen Nachbarn, die die Idee hatten, eine umfassende Karte von Bayern zu erstellen.*
**Beam search BLEU/log-likelihood:** 0.400/-12.754

## C.3 Different choice of hyperparameter $\alpha$ for the Dirichlet prior

For the Dirichlet prior, indeed $\alpha$ has a meaningful impact on the performance of ULTS: It can be seen as a hyperparameter that trades exploitation for exploitation (smaller value means more exploration).

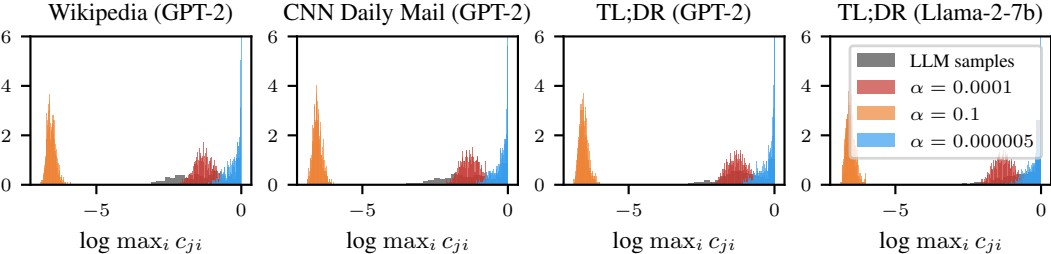

**Figure 8:** Distribution of the maximum of categorical distributions sampled from an LLM, as well as from Dirichlet priors with different concentration parameters.

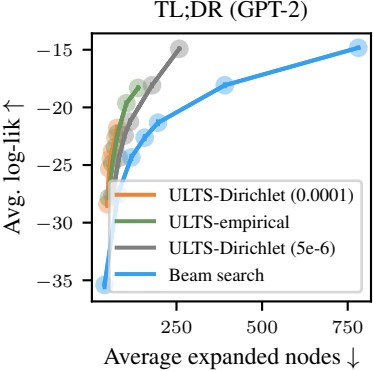

**Figure 9:** Comparison of two different values $\alpha = 0.0001$ and $\alpha = 5e - 6$ of the Dirichlet prior concentration parameter in ULTS

So, the choice of $\alpha$ should be based on how much budget one has since more exploration means more node expansions. Figure 9 shows a comparison of two different values $\alpha = 0.0001$ and $\alpha = 5e - 6$.

### C.4 NUMBER OF SAMPLES IN ULTS

Table 2 shows results for an ablation for hyperparameter $N$ on 100 sentences from the machine translation task with maximum beam size $k_{max} = 4$. Only for $N = 1$ the performance is slightly worse, indicating that a small number like $N = 10$ might already be sufficient.

| $N$ | Avg. log-lik | node expansions | Time in s |
|---|---|---|---|
| 1 | -9.454 | 86.64 | $3.488 \pm 0.187$ |
| 10 | -9.066 | 90.75 | $3.631 \pm 0.208$ |
| 100 | -9.097 | 92.28 | $3.690 \pm 0.211$ |
| 1000 | -9.074 | 92.41 | $3.718 \pm 0.213$ |

**Table 2:** Results for different choice of number of samples $N$.

### C.5 ADDITIONAL RUNTIME RESULTS

In Table 3 we show runtime results for beam search with beam size $k = 5$ and ULTS with maximal beam size $k_{max} = 3$ and $k_{max} = 5$. Despite expanding fewer nodes than beam search, ULTS is currently slower in settings where different nodes expansion in beam search can be batched. However, note that batching is not always possible, e.g. in memory-constrained settings (the memory resources depend on the model size, sequence length, as well as batch size).

| Method | $k$ or $k_{\max}$ | Time in s |
|---|---|---|
| Beam search | 5 | $2.158 \pm 0.029$ |
| ULTS Dirichlet | 3 | $4.622 \pm 0.114$ |
| ULTS Dirichlet | 5 | $5.177 \pm 0.164$ |

**Table 3:** Total runtime for decoding one of the TL;DR input prompts with Llama-2-7b.

As a reference regarding the computation of the prior: Building the Dirichlet Prior for a tree of depth 250 an branching size 32256 with 1000 samples on a desktop machine (MacBook M1) with CPU only takes 10:50 min (2.6 secs per level of the tree).

### C.6 ALTERNATIVE ACQUISITION FUNCTION

As discussed in Section 3.4, different selection (and hence backup) strategies can be utilized. Recall that all of our results so far are obtained using the "posterior descendant" strategy defined in (2). Here, we show the corresponding results where the other strategy ("posterior") is used instead.

First, Fig. 10 shows results under the same setting as in the main text, but both ULTS-Dirichlet and ULTS-Emp use the "posterior" strategy instead. We noticed that this strategy also performs well—it is more efficient than beam search. Moreover, it also achieves better or similar likelihood than beam search in Llama-2-7b.

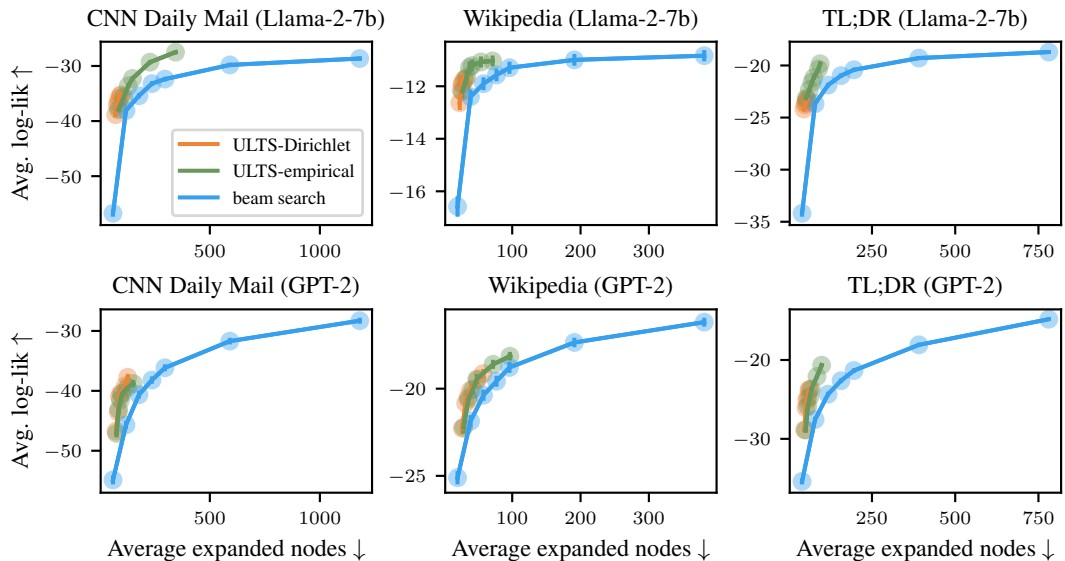

**Figure 10:** Decoding experiments with Llama-2-7b and GPT-2 with the "posterior" strategy.

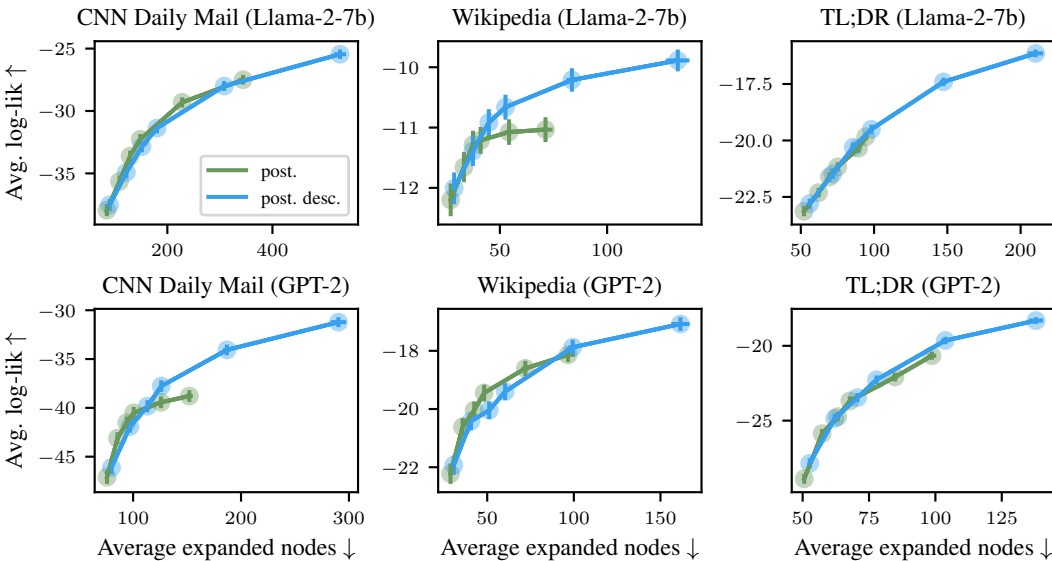

**Figure 11:** "Posterior" vs. "posterior descendant" acquisition function.

We further compare the "posterior" strategy compared to the "posterior descendant" strategy in Fig. 11. We notice that the "posterior" strategy tends to underexplore compared to the "posterior descendant" strategy. Hence, we use and recommend the "posterior descendant" strategy by default.

## C.7 FULL RESULTS FOR DECODING EXPERIMENTS WITH LLAMA-2-7B AND GPT-2

Figure 12 shows the full results from the decoding experiments with LLama-2-7b and GPT-2 from section 5 in the main text.

## C.8 DETAILS FOR EXPERIMENT WITH REPETITION PENALTY

In this task, we complete 100 sentences from the Wikipedia dataset with GPT-2. We predict 100 tokens. We define the diversity term $b(x)$ of a node $x$ based on the proportion of duplicated $n$-grams in the token sequence $(a_1, ..., a_i)$ corresponding to the node. For a token sequence $(a_1, ..., a_i)$, let

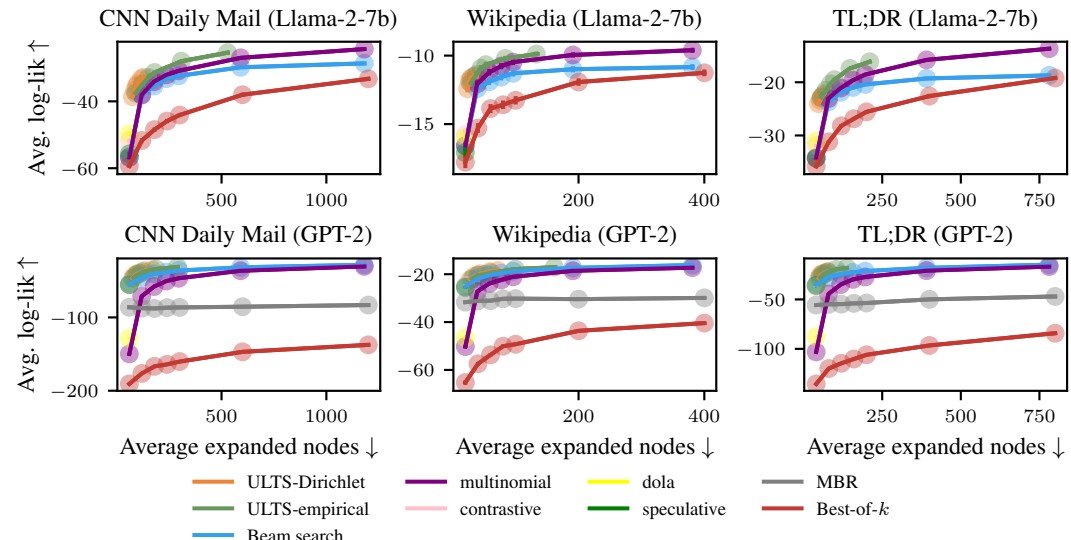

**Figure 12:** Full results for decoding experiments with LLama-2-7b and GPT-2

| Method | Perplexity ↓ | Diversity ↑ |
|---|---|---|
| Beam search | 1.16±0.01 | 0.33±0.01 |
| Nucleus sampling | 3.69±0.07 | 0.82±0.01 |
| Contrastive search | 1.27±0.01 | 0.4±0.01 |
| ULTS | 1.16±0.01 | 0.33±0.01 |
| ULTS with $\lambda = 0.2$ | 1.3±0.01 | 0.55±0.01 |
| ULTS with $\lambda = 0.4$ | 1.45±0.01 | 0.74±0.01 |
| ULTS with $\lambda = 0.6$ | 1.5±0.01 | 0.79±0.01 |
| ULTS with $\lambda = 0.8$ | 1.53±0.01 | 0.82±0.01 |
| ULTS with $\lambda = 1.0$ | 1.55±0.01 | 0.83±0.01 |

**Table 4:** Effects of a penalty term in ULTS' acquisition function.

$\textbf{rep-n}((a_1, ..., a_i)) = \left(1 - \frac{|\text{unique } n\text{-grams}((a_1,...,a_i))|}{|\text{total } n\text{-grams}((a_1,...,a_i))|}\right)$. Following Su et al. (2022), we measure diversity by taking repetitions at different $n$-gram levels into account:

$$\textbf{diversity}((a_1, ..., a_i)) = \prod_{n=2}^{4} (1 - \textbf{rep-n}((a_1, ..., a_i))).$$

Since we ULTS optimizes the total likelihood of a sequence and not the average likelihood of a sequence, we additionally scale the term with the tree depth, leading to $b(x) = d \cdot \log \textbf{diversity}((a_1, ..., a_i))$ in log-space. For ULTS we use a Dirichlet prior with concentration parameter $\alpha = 5 \times 10^{-6}$, $\epsilon = 0.1$, $k_{max} = 5$. and penalty parameter $\lambda \in \{0.0, 0.2, 0.4, 0.6, 0.8, 1.0\}$. The parameters for contrastive search are top-$k = 5$ and penalty parameter $\alpha = 0.6$. For beam search, we use $k = 5$. The main paper reports results for ULTS with $\lambda = 0.0$, i.w. without penalty, and $\lambda = 0.2$. The full results are given in Table 3. One can see that with increasing penalty the diversity of the decoded sequences increases, i.e. extending the acquisition function with a repetition term is effective.

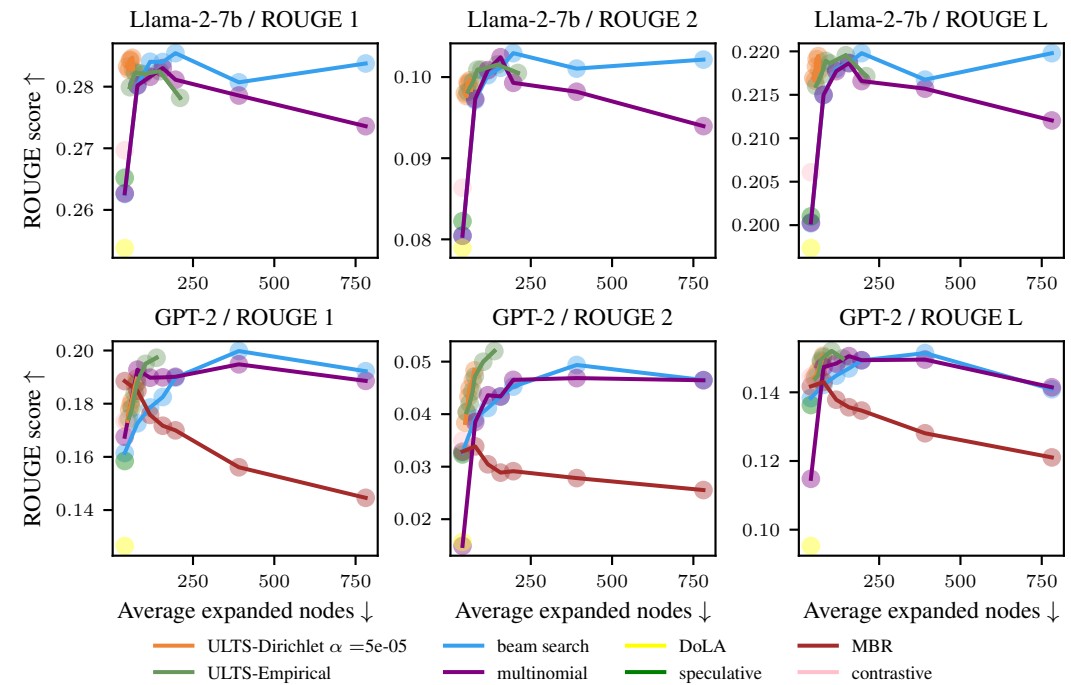

**Figure 13:** ROUGE scores for the summarization task.

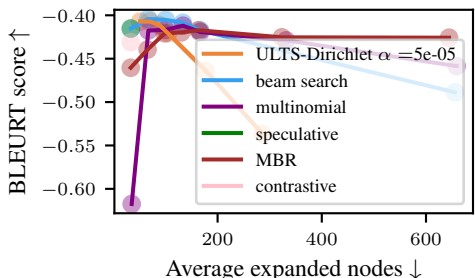

**Figure 14:** BLEURT scores for the machine translation task.

## C.9 Additional metrics

Figure 13 shows an evaluation in terms of ROUGE metrics for the summarization task on the TL;DR dataset described in Section 5. ROUGE 1 quantifies the overlap of unigrams between the produced sentence and ground truth reference, ROUGE 2 refers to bigrams, and ROUGE L refers to the longest commen subsequence. Though there is no perfect correlation between the ROUGE metric and the log-likelihood, ULTS still tends to achieve a good trade-off between performance and node expansions overall. Figure 14 and 15 show additional results in terms of the BLEURT and COMET metrics. Here, ULTS performs well for small $k_{max}$ but shows a decrease in performance for larger $k_{max}$.

## C.10 ULTS with larger budget

Fig. 17 shows additional results for ULTS with the empirical prior and GPT-2 on the Wikipedia dataset, as well as ULTS with the Dirichlet prior and t5 on the machine translation task.

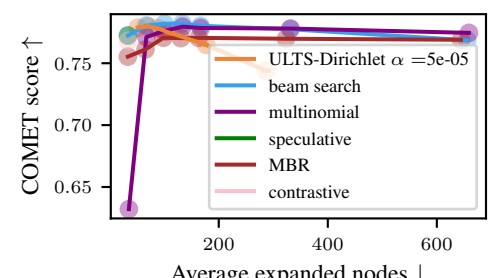

**Figure 15:** COMET scores for the machine translation task.

# APPENDIX D PSEUDOCODE

---

**Algorithm 1:** Prior belief over $\Delta$.

---

**Input:** Depth of the tree $d$, branching size of tree $b$, prior $p(\boldsymbol{c})$ over LLM softmax outputs, number of samples used for the approximation $N$

**Output:** Table of parameters $\text{params}_l$ for Beta distributions $\mathcal{B}_l$ for each level $l$ of the tree

**for** $l \leftarrow d - 1$ **to** $1$ **do**

    **for** $n \leftarrow 1$ **to** $N$ **do**

        $c_n \sim p(\boldsymbol{c})$

        **for** $j \leftarrow 1$ **to** $b$ **do**

            **if** $l = d - 1$ **then**

                $\Delta_{nj} \leftarrow 1$

            **else**

                $\Delta_{nj} \sim \mathcal{B}_{l+1}(\Delta \mid \text{params}_{l+1})$

            **end**

        **end**

        $\Delta_n \leftarrow \max_{j=1,\ldots,b}(c_{nj} \cdot \Delta_{nj})$

    **end**

    $\text{params}_l \leftarrow \texttt{beta-MLE}(\{\Delta_n\}_{n=1}^N)$

**end**

---

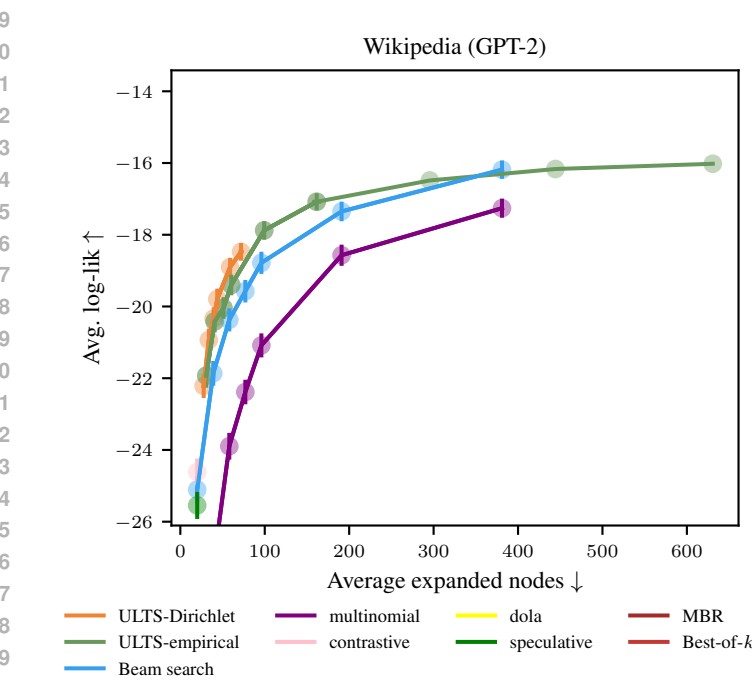

**Figure 16:** Comparison of ULTS with the baselines on the Wikipedia dataset as described in Section 5 with additional data points for $k_{\max} \in \{50, 100, 200\}$.

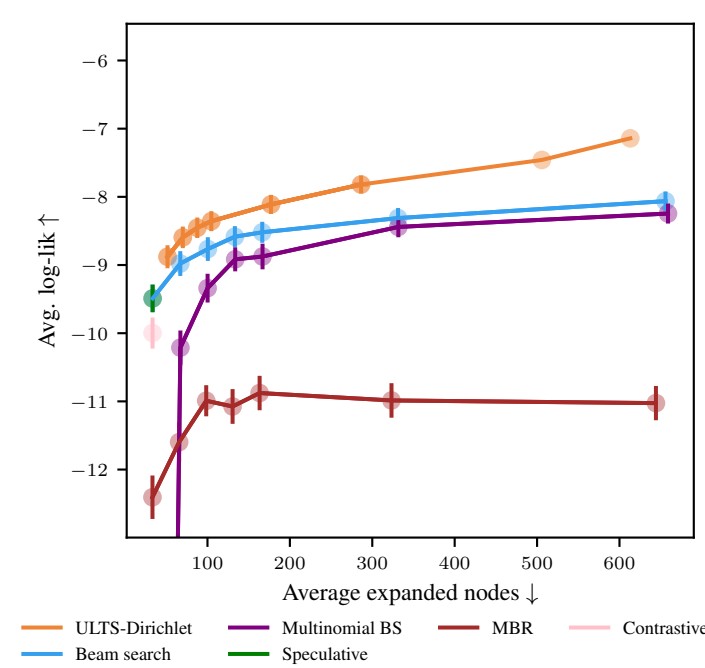

**Figure 17:** Comparison of ULTS in the machine translation task as described in Section 5 with additional data points for $k_{\max} \in \{50, 100\}$.

**Algorithm 2:** ULTS.

---

**Input:** Num. of tokens to generate $d$, number of samples $N$, approximate priors $\{\mathcal{B}_l\}_{l=1}^{d}$,
$\qquad$ confidence parameter $\varepsilon$, maximum number of expandable nodes per level $k_{\max}$
**Output:** A leaf $x^*$ and its likelihood $c^*$
// Initialization
$\mathcal{L} \leftarrow \{\boldsymbol{x}_0\}$
**for** $n \in 1, ..., N$ **do**
$\quad | \quad (v_{\boldsymbol{x}_0})_n \sim \mathcal{B}_1(\Delta)$
**end**
$c^*, x^* \leftarrow -\infty, \text{None}$
**while** $\hat{\mathbb{P}}(c^* < v_{\boldsymbol{x}_0}) > \varepsilon$ **do**
$\quad$ // Selection always starts from root
$\quad x_i \leftarrow \texttt{select}(\boldsymbol{x}_0, k_{\max})$
$\quad$ // Expand
$\quad \mathcal{L} \leftarrow (\mathcal{L} \setminus \{x_i\}) \cup \text{children}(x_i)$
$\quad$ **for** $x_c \in \text{children}(x_i)$ **do**
$\quad\quad$ // Generate posterior samples
$\quad\quad$ **for** $n \in 1, ..., N$ **do**
$\quad\quad\quad | \quad (\Delta_{x_c})_n \sim \mathcal{B}_{\text{level}(x_c)}(\Delta)$
$\quad\quad\quad | \quad (v_{x_c})_n \leftarrow c_{\boldsymbol{x}_0 \to x_c} \cdot (\Delta_{x_c})_n$
$\quad\quad$ **end**
$\quad\quad$ // Update best complete path so far
$\quad\quad$ **if** $\text{level}(x_c) = d$ **and** $c_{\boldsymbol{x}_0 \to x_c} > c^*$ **then**
$\quad\quad\quad | \quad c^* \leftarrow c_{\boldsymbol{x}_0 \to x_c};\ x^* \leftarrow x_c$
$\quad\quad$ **end**
$\quad$ **end**
$\quad \texttt{backup}(\{(v_{x_c})_n\}_c, \boldsymbol{x}_0 \to x_c)$
$\quad$ // termination probability
$\quad \hat{\mathbb{P}}(c^* < v_{\boldsymbol{x}_0}) \leftarrow \frac{1}{N} \sum_{n=1}^{N} \mathbb{I}[c^* \leq (v_{\boldsymbol{x}_0})_n]$
**end**

---

