# OpenReview forum: "Uncertainty-Guided Optimization on Large Language Model Search Trees"
_ICLR.cc/2025/Conference — Submitted to ICLR 2025_

### Official Review · Reviewer_iHdP · 2024-10-24

**Soundness:** 3
**Presentation:** 3
**Contribution:** 3
**Rating:** 8
**Confidence:** 3

**Summary:**

The paper proposes a novel non-myopic decoding strategy for LLMs.
They phrase the problem as Bayesian optimization, which lets them alleviate the expensive expansion of the search tree, that plagues other non-myopic tree search algorithms such as MCTS.
This is done by means of a pre-computed prior, which is updated by the evidence gathered in the decoding process.
Experiments demonstrate higher likelihood of decoded sequences and lower runtimes compared to baselines, with the caveat that batching for further efficiency has not been demonstrated yet.

**Strengths:**

Decoding is at the heart of improving LLMs, thus a better decoding strategy, even if it is only applicable in some specific cases, might have huge impact on the field.
The proposed algorithm performing bayesian optimization appears very natural in the setting and provides a way for non-myopic decoding using beliefs about future token probabilities or potentially other quantities such as harmfulness etc.
The formulation of the algorithm and its contextualization in prior work are very strong and the writing is of very high quality in this sections.

**Weaknesses:**

I think in its current state, the experimental evaluation, especially how it is presented, but also some details of experiments are the main weakness of the paper. Details as follows.

### Major
* I don't understand the experiments presented in section 5.2, e.g. where do I find the results for the generation and where the results for the summarization task?
* While hyperparameters and details for ULTS are well explained in the experimental setup, the baselines are hardly explained.
Even if standard values of the huggingface library are used, please state them at least in the appendix to make the experiment reproducible if the standard values in the library change eventually.

### Minor
* The captions from Figure 3 onwards are very short and could convey more information about the depicted experiment.
They could be more self-contained, otherwise I have to find where in the text the figure is explained.
* The y-range in Figure 3 is a bit odd (for the left plot), I would extend it a bit to include the contrastive baseline.
* I was puzzled about the main results (Fig. 3 and 4) for a long time, until I figures out that different dots correspond to different values of k/k_{max}.
This should be made more obvious in the description of the results and in the figure caption.
* I don't find an ablation over the number of samples N used for the approximation (c.f. Alg 2). I understand this is not the costly part of the algorithm, but it would be nevertheless interesting for a practitioner to understand whether this parameter has a lot of influence.

Very generally, the quality of writing differs a lot between the experimental section (and some sections in the appendix, e.g. C.4) and the rest of the paper.

### Grammar, etc.
* line 343/344 wrong citation style after contrastive search
* line 366 "... are done on **a** single ..."
* line 367 "... in the tree **if** the <EOS> ..."
* !! line 369 "The results are in **Figure 3**. ULTS **is** both ..."
* line 403 "We use **a** context length ..."
* line 444 "**This is** outside of the present work's scope, but is a promising **direction for** future work"
* line 841 "... currently **slower in** settings..."
* line 843 "... in **memory-constrained** settings ..."

**Questions:**

* I have to a-priori specify a tree depth d for ULTS, but can I just select a rather large one to be sure that I can generate a good answer, or do I have to exactly guess the length of the answer I would like to have?
Obviously, if the best answer is longer than the specified depth, the algorithm doesn't work, but what about the other case, that it is (much) shorter than the specified length?
* What models are used for the speculative decoding baseline? Is the generating model the same as the one used for the other baselines? What is the second model?
* In line 364, what is referenced by "... the strategy in **(2)** for the selection ..."? Is equation (2) meant? The same goes for line 850 in the appendix.
* Why don't you filter the test datasets as described in section 5.2. before randomly selecting a subset?
* What is the rationale behind the different output sequence lengths for the different datasets in section 5.2?
Was this somehow chosen systematically prior to the experiments?
As described in the paper, it appears arbitrary and oddly specific to do it differently for the different datasets.
* Where do I see the summarization task described in lines 409 - 412?
* I understand that computing the prior can be done once, but I would nevertheless be interested in how long it takes roughly for certain depth and width of the search tree, to understand to which scenarios we can expect the algorithm to be useful. E.g. if we wait 100 years to compute a prior for depth 256 and width 256k on 10,000 samples, it is likely not usable in many applications. This would be important to know, even if only stated in the appendix.

---

> ### Author Response · Authors · 2024-11-20
>
> **[1/2]**
>
> Thank you for your detailed comments! Please have a look at the revised version, where we included your suggestions for improved presentation.
>
> >I don't understand the experiments presented in section 5.2, e.g. where do I find the results for the generation and where the results for the summarization task?
>
> Both results are in Figure 4. The results from the generation task are in the first two columns ("CNN"/"Wikipedia") and the results from the summarization task are in the last column ("TL;DR"). We expanded the figure caption to make this clearer.
>
> > While hyperparameters and details for ULTS are well explained in the experimental setup, the baselines are hardly explained. Even if standard values of the huggingface library are used, please state them at least in the appendix to make the experiment reproducible if the standard values in the library change eventually.
>
> We added this information to the Appendix, Section B.
>
> > I don't find an ablation over the number of samples N used for the approximation (c.f. Alg 2). I understand this is not the costly part of the algorithm, but it would be nevertheless interesting for a practitioner to understand whether this parameter has a lot of influence.
>
> We added an ablation in Appendix C.4, Table 2, with `N = 1, 10, 100, 1000`. We found that ULTS performs well, even with a small number of samples like `N = 10`.  Even `N = 1` is actually already quite good as it coincides with a hierarchical version of Thompson sampling.
>
> | Num samples | Loglike | Node expansions | Avg. wallclock (s) |
> |-------------|---------|-----------------|--------------------|
> | 1           | -9.454  | 86.64           | 3.488±0.187        |
> | 10          | -9.066  | 90.75           | 3.631±0.208        |
> | 100         | -9.097  | 92.28           | 3.690±0.211        |
> | 1000        | -9.074  | 92.41           | 3.718±0.213        |
>
>
> > I have to a-priori specify a tree depth d for ULTS, but can I just select a rather large one to be sure that I can generate a good answer, or do I have to exactly guess the length of the answer I would like to have? Obviously, if the best answer is longer than the specified depth, the algorithm doesn't work, but what about the other case, that it is (much) shorter than the specified length?
>
> Yes, that's what we did in the machine translation experiment. In the context of the other reviews, we also added another test setting with a maximum depth of 500 on the HumanEval dataset, and we found ULTS works well in this case. The maximum depth we set for ULTS corresponds to the maximum sequence length one would also fix for other standard decoding methods like beam search. A follow-up idea would be to infer a distribution of sentence lengths and then marginalize the sentence length when computing the acquisition function.
>
> > What models are used for the speculative decoding baseline? Is the generating model the same as the one used for the other baselines? What is the second model?
>
> The generating model is the same as the other baselines. The second model is the one specified by HuggingFace as the default. We only set `​​prompt_lookup_num_tokens` (to `10`) as suggested here: <https://huggingface.co/docs/transformers/en/main_classes/text_generation>. So, the second model is the n-gram model. We emphasize that speculative decoding retains the performance of the base decoding method (greedy) and focuses only on speeding up the decoding process.
>
>
> > In line 364, what is referenced by "... the strategy in (2) for the selection ..."? Is equation (2) meant? The same goes for line 850 in the appendix.
>
> Yes. We edited it accordingly.
>
> > Why don't you filter the test datasets as described in section 5.2. before randomly selecting a subset?
>
> For the summarization experiment with TL;DR dataset,  the input prompt consists of the full text, and the task is to summarize it. Thus, this is not a text-continuation problem and therefore we do not need to filter the contexts. For the other datasets, we do filtering because we want to make sure that we are not close to the end of the text sample so that it still makes sense to generate more tokens.
>
> > What is the rationale behind the different output sequence lengths for the different datasets in section 5.2? Was this somehow chosen systematically prior to the experiments? As described in the paper, it appears arbitrary and oddly specific to do it differently for the different datasets.
>
> We did so to provide an additional level of ablation to our method. Moreover, during this rebuttal, we also added a new experiment with a maximum depth of 500 (HumanEval). ULTS also works well in this case.

---

> ### Author Response · Authors · 2024-11-20
>
> **[2/2]**
>
> > I understand that computing the prior can be done once, but I would nevertheless be interested in how long it takes roughly for certain depth and width of the search tree, to understand to which scenarios we can expect the algorithm to be useful. E.g. if we wait 100 years to compute a prior for depth 256 and width 256k on 10,000 samples, it is likely not usable in many applications. This would be important to know, even if only stated in the appendix.
>
> We added wall-clock results for this to the appendix, Section C.5. On our consumer-grade machine
> (MacBook M1) with CPU only, it takes 10:50 mins to compute the Dirichlet prior for a tree of depth 250 and branching size 32256 with 1000 samples (2.6 secs per level).

---

> > ### Comment · Reviewer_iHdP · 2024-11-22
> >
> > Thank you for your response. Many of my concerns have been addressed.
> >
> > The new HumanEval tasks adds very positive to the strength of the empirical evaluation.
> > However, it also raises new questions:
> > * Why is Beamsearch-Mult basically as fast as UTLS ($k_{max} = 10$) while having 2.5 times as many node expansions?
> > * Why does UTLS ($k_{max} = 20$) have fewer node expansions than UTLS ($k_{max} = 10$)?
> > * Generally the results for UTLS ($k_{max} = 20$) looks strange, the LL is highest, yet the pass@1 pretty low, why is that?
> >
> > Furthermore, I would also like to see the utility of ULTS at a higher number of expanded nodes for experiments in the now figure 5.
> > Reviewer UmM7 is indeed correct that with the current experiments we do not know whether ULTS is at the pareto-frontier, it could also be worse than other sampling methods for a high number of node expansions.

---

> > > ### Author Response · Authors · 2024-11-22
> > >
> > > Thank you very much for your prompt reply and engagement! We are happy to hear that many of your concerns have been addressed and that our rebuttal adds very positively to the strength of our evaluation. We hope that you reflect this in your final score.
> > >
> > > > Why is Beamsearch-Mult basically as fast as UTLS ($k_\max = 10$) while having 2.5 times as many node expansions?
> > >
> > > This is due to the fact that we do not use batch expansion. We have mentioned this as future work in our original submission since this requires a specialized acquisition function as in batch Bayesian optimization literature. See also our response to **Reviewer 8FSd**.
> > >
> > > > Why does UTLS ($k_{max} = 20$) have fewer node expansions than UTLS ($k_{max} = 10$)?
> > >
> > > It can happen that there are nodes where the `<EOS>` token is within the top-20, but not within the top-10. Note that this phenomenon is not limited to ULTS: It has been observed in beam search, higher $k$ produces shorter sequences (see  Fig. 9 of [1]).
> > >
> > > > Generally the results for UTLS ($k_{max} = 20$) looks strange, the LL is highest, yet the pass@1 pretty low, why is that?
> > >
> > > This is because due to the inherent miscalibration of LLMs: higher likelihood does not perfectly correlate with task-specific qualitative metrics. We already discussed this in our “Limitations” in Sec. 3.5 of our paper.
> > >
> > > > I would also like to see the utility of ULTS at a higher number of expanded nodes for experiments in the now figure 5
> > >
> > > As we also mentioned to **Reviewer UmM7**, please find new results with a larger $k_{max}$ for ULTS on Wikipedia with GPT2 and WMT-19 with T5 in Figure 14 and Figure 15 in Appendix C.10. We see that ULTS does indeed lie at the Pareto frontier in both cases. On the wikipedia dataset beam search eventually catches up.
> > > We will add results with higher node expansions for other dataset-LLM combinations, too. Thanks for the suggestion.
> > >
> > > Importantly, the results we have already encompass a realistic range of computational budgets, providing evidence that ULTS will be effective in practical applications. For instance, beam search with \(k = 5\) or \(k = 10\) represents a plausible choice (see also [1]) and one can already see that ULTS reaches this performance level.
> > >
> > >
> > > **References**
> > >
> > > [1] Gian Wiher, Clara Meister, and Ryan Cotterell. On decoding strategies for neural text generators. TACL, 10, 2022

---

> > > > ### Comment · Reviewer_iHdP · 2024-11-25
> > > >
> > > > Thank you for the additional clarifications. Together with the original rebuttal, they improved my perception of this work. I support accepting this paper and have updated my score accordingly.

---

> > > > > ### Author Response · Authors · 2024-11-25
> > > > >
> > > > > Thanks again for your time and for your support!

---

### Official Review · Reviewer_UmM7 · 2024-10-27

**Soundness:** 3
**Presentation:** 2
**Contribution:** 3
**Rating:** 6
**Confidence:** 4

**Summary:**

This paper proposes a probabilistic approach of sampling from language models. The softmax probabilities in each step are modeled as independent random variables with a chosen (dirichlet) or an estimated (empirical) prior. Then, according to the probabilistic model, a method of choosing the next token is proposed, which requires a precomputed prior of the optimal gain and an estimated acquisition function. The work then discusses limitations and related works of the idea.

In the experiments, it is shown that the proposed approach, named ULTS, gives higher probabilities with fewer "averaged expanded nodes" across different datasets. Also, the additional runtime overhead from ULTS is small comparing to the decoding part of LLMs. The experiments also explore a different acquisition function which trades perplexity for diversity.

**Strengths:**

Overall, the presentation of the experimental results is good and many related works are adequately discussed. The batch acquisition strategy discussed in the conclusion section looks interesting and may be useful in the future.

**Weaknesses:**

I am from the Bayesian side but I do not buy the story of Bayes in this paper.
- Section 3.3 is on posterior beliefs of the optimal values. The probabilistic model is so strange that the posterior becomes the same as the prior for $\Delta_i$. It only tells me that the observation, which is the path from $\textbf{x}_0$ to $x_i$, gives no information about the future. This makes sense given its independent assumption. However, from my view, it is simply a heuristic decoding algorithm without any Bayes in it.
- The work compares itself with Bayesian optimization techniques multiple times (line 39, line 260). But I still do not see how the "Bayesian" comes in. The only part that may be related is the backup procedure in line 262-269. However, the term "propagate" in this part seems crucial but is never clearly defined. How will the "prior" or the "acquisition function" be updated after an action? Also, the "backup" function in the pseudocode of Algorithm 2 is never defined.
- The work uses the Beta distribution to bound the probabilities in the unit interval, claiming that it is beneficial. However, there is no evidences showing why it is the case.
- The work states that the approach is non-myopic, so it can acquire higher probability sequences in the decoding procedure. However, it is not supported in the experiments, especially in Figure 4. If I do not care about the number of expanded nodes, ULTS does not seem to produce sequences with higher probabilities, which really weakens the statement.
- I would also argue that the i.i.d. assumption is too strong, since it ignores the context in a decoding step. I think at least the context is important to be a probabilistic approach in decoding.

Minors:
- In line 144, there is a definition of increment for games, which may be confusing when looking att other parts of the paper where $\Delta_i$ is defined differently.

I also have some concerns about the experiments, which are detailed in the next part.

**Questions:**

- In the experiments, how are the "average expanded nodes" defined for different approaches? Algorithm 2 looks at every children, (and possibly every grandchildren taking equation (2)). Does that mean ULTS may expand $b^2$ nodes each step?
- Where is $k_{\max}$ in Algorithm 2? A novel algorithm is proposed but the key hyperparameter to control its complexity should at least be in its pseudocode.
- Following the weaknesses, how is the "backup" function implemented? Is it a Bayesian update of the prior or the acquisition function?

---

> ### Author Response · Authors · 2024-11-20
>
> Thank you for your review! Here we discuss your comments individually. We have updated the paper to address your comments and other reviewers’ comments.
>
> If you have further comments, we would be very happy to discuss them with you!
>
>
> > Section 3.3 is on posterior beliefs of the optimal values. The probabilistic model is so strange that the posterior becomes the same as the prior for $\Delta_i$. It only tells me that the observation, which is the path from $\textbf{x}_0$ to $x_i$, gives no information about the future. This makes sense given its independent assumption. However, from my view, it is simply a heuristic decoding algorithm without any Bayes in it.
> > The work compares itself with Bayesian optimization techniques multiple times (line 39, line 260). But I still do not see how the "Bayesian" comes in. The only part that may be related is the backup procedure in line 262-269. However, the term "propagate" in this part seems crucial but is never clearly defined. How will the "prior" or the "acquisition function" be updated after an action?
>
> The principle of Bayesian optimization is to leverage sequential decision-making under uncertainty. I.e., given an unknown variable relevant to decision-making process, we put a prior belief on the unknown, update it with our observation, and pick an action that maximizes the expected utility under the a posteriori belief.
>
> In ULTS’ case, the actions are the nodes to expand, and the unknown of interest is the _value_ of each node, which tells us “if I pick this node, what is the reward (total likelihood) that I will get if I go down the tree optimally?”. Hence, we put prior beliefs over values of the nodes in the tree by backing up prior samples (Alg. 2; i.e. recursively taking maximums in a bottom-up fashion). Then, during the decoding process itself, as we expand the tree (i.e. querying the LLM), we update the belief at the frontier node. Probabilistically speaking, it makes sense that a posterior sample at a frontier node factorizes into the observation term and the prior-sample term since, at this point, ULTS does not have any observation about the future, i.e. further down the tree.
>
> However, we would like to point out that this only happens at the frontier nodes. The nodes that we have expanded previously, up until the root node, will also be updated due to our “Backup” step. Thus, in the next “Selection” round, we select a node recursively, starting from the root, based on the posterior belief of the interior node that takes into account the future up to the frontier node.
>
> Note that the acquisition function during this step is computed by approximating the expected utility under such a belief, akin to the principle of sequential decision-making under uncertainty above.
>
> Please refer to the newly added Figure 2 in our paper for an illustration.
>
> In any case, you are right that one can see ULTS as a heuristic best-first search, as we have mentioned in the Related Work section. However, ULTS is unlike the standard heuristic search like A* due to the interpretation above, especially since ULTS also performs exploration since the decision rule is influenced by the a posteriori uncertainty over values.
>
> We have clarified this in the paper. If you have further comments and suggestions, we would be happy to address them!
>
>
> > Definition of the backup function.
>
> We added the corresponding formulas, see equations (5) and (6), to the descriptions of the backup functions. Note that we experimented with two different alternatives for the backup operator. The simplest of which is to propagate the posterior samples upward by recursively taking their maximums as in the computation of the priors (Algorithm 1). Please see also the newly added Figure 2 for illustration.
>
>
> > The work uses the Beta distribution to bound the probabilities in the unit interval, claiming that it is beneficial. However, there is no evidences showing why it is the case.
>
> Recall that our beliefs are over values, which have the same properties as probabilities since the rewards of the tree are the softmax outputs of the LLM. The values are computed by recursively taking the maximum of probabilities starting from the leaves, so this maximum can also be interpreted as a probability. This implies that the Beta distribution is a good choice for modeling them since it has support on $[0, 1]$.
>
> In any case, we reemphasize that the choice of the Beta distribution is simply to amortize the cost of storing the prior. Without it, we would need to store many prior samples over values. With this amortization scheme, we only need to store Beta parameters for each level of the tree.

---

> ### Author Response · Authors · 2024-11-20
>
> **[2/2]**
>
> > The work states that the approach is non-myopic, so it can acquire higher probability sequences in the decoding procedure. However, it is not supported in the experiments, especially in Figure 4. If I do not care about the number of expanded nodes, ULTS does not seem to produce sequences with higher probabilities, which really weakens the statement.
>
> We would like to re-emphasize that ULTS targets two objectives: maximizing probability/likelihood _and_ minimizing the number of node expansions (fewer LLM queries). Note that these objectives are also the objectives of general Bayesian optimization: one wants to maximize an unknown function while querying as few data points as possible.
>
> Our experiments reflect these: The plots in our Experiment section are there to show that ULTS, on average, lies at the **Pareto frontier** of these objectives, thus dominating other baseline methods. We expanded the figures’ captions accordingly.
>
>
> > I would also argue that the i.i.d. assumption is too strong, since it ignores the context in a decoding step. I think at least the context is important to be a probabilistic approach in decoding.
>
> This is a great idea. While in this paper we have shown that ULTS performs well across different test cases (see also an additional code-generation test case with the HumanEval dataset), your idea of extending ULTS to take into account the context can be a interesting future work.
>
> Note however that, in this case, ULTS’ priors cannot be shared for different decoding runs since it will depend on the context. One might be able to amortize this by parametrizing the prior with a small LLM, though. However, this will lead to a more expensive decoding method.
>
>
> > In line 144, there is a definition of increment for games, which may be confusing when looking at other parts of the paper where $\Delta_i$ is defined differently.
>
> Yes, the definition of $\Delta_i$ is different in our case since our setting/objective is different from the game tree setting in the related work. We removed the corresponding definition of how  $\Delta_i$ is defined in the related work in order to prevent potential confusion of the reader.
>
>
>
> >In the experiments, how are the "average expanded nodes" defined for different approaches? Algorithm 2 looks at every children, (and possibly every grandchildren taking equation (2)). Does that mean ULTS may expand $b^2$ nodes each step?
>
> The number of node expansions corresponds to the number of forward passes through the LLM (without batching). In each decoding run (i.e., for each context), we track/calculate this number and we report the average of these numbers across different contexts in a given dataset.
>
> A single node expansion results in the observation of a single LLM’s softmax output (a probability vector) with $b$ components. Thus, only $b$ children are added to the tree.
>
> While equation (2) looks complicated, the recursive part is done during the “Backup” step. During the “Selection” step, we simply use the backed-up posterior samples to make a decision in eq. (1) without any recursive part. We hope that the newly added example in Figure 2 also helps to clarify this.
>
> > Where is $k_{\max}$ in Algorithm 2? A novel algorithm is proposed but the key hyperparameter to control its complexity should at least be in its pseudocode.
>
> $k_\text{\max}$ is hard constraint that ensures that no more than $k_{\max}$ nodes per level are selected during the decoding run. We have made this explicit in see Algorithm 2 in the revised version. Please note that we had to move Algorithm 2 to the Appendix due space limitations.
>
> > Following the weaknesses, how is the "backup" function implemented? Is it a Bayesian update of the prior or the acquisition function?
>
> Please refer to our answer to your first “Weakness” above. Essentially, our backup can be done by following Alg. 1 (taking maximum over samples recursively from the frontier node up until the root), but by replacing the prior with the probability obtained from an LLM call. Other backup operators can also be considered.
>
> Please also refer to the newly added Figure 2 for illustration and also equations (5) and (6).

---

> ### Comment · Reviewer_UmM7 · 2024-11-21
>
> I've spent some time looking at the revision. It makes a bit more sense to me now. However my main concerns are still not resolved.
>
> So, if I am understanding it correctly, the logic of the work is as follows. There is a prior belief of what remaining values at each level could have. Instead of searching until a leaf, ULTS stops at an unexpanded node, and use the prior belief to update the utility function of the tree search, in order to make the search more efficient. Also, there is a prior belief of how the softmax outputs are distributed, which controls the exploitation-exploration tradeoff. The add-ons to MCTS are prior sampling instead of posterior sampling. To argue that the ULTS is Bayesian, it is equivalent to argue that MCTS alone is Bayesian. I still keep my opinion that the proposed method does not look like Bayesian optimization. A simple Google search in the literature leads to [1], which is in a similar topic, but they at least use GP and the unobserved are sampled from a posterior. Note that their search is not dependent on predefined priors, but it makes the method depend on the context.
>
> The other main concern is about the experiments (in the now Figure 5). The authors argue that ULTS lies at the Pareto frontier of all methods. However, nothing prevents ULTS to be tested with higher "average expanded nodes". Look at the Wikipedia dataset, I hardly believe that beam search finds the optimal decodings with 400 expanded nodes. So if ULTS is really useful, it should be able to find better decodings with 400 expanded nodes.
>
> The other responses make me more favorable of this work. But I am still not swayed to change my score.
>
> [1] Mern, J., Yildiz, A., Sunberg, Z., Mukerji, T., & Kochenderfer, M. J. (2021, May). Bayesian optimized monte carlo planning. In Proceedings of the AAAI Conference on Artificial Intelligence (Vol. 35, No. 13, pp. 11880-11887).

---

> > ### Author Response · Authors · 2024-11-22
> >
> > Thanks so much for your quick reply and engagement! We are glad to hear that our rebuttal answered most of your questions and you are now more in favor of our work.
> > We hope that you reflect this in your final score.
> >
> > You are correct regarding the prior over remaining values at each level and that ULTS relies on this prior to guide search non-myopically. However, the expected utility (acquisition function) for frontier nodes is computed via the combination of both observations (the LLM outputs, via the dirac likelihood) and the samples from the aforementioned prior, hence the posterior interpretation (Fig. 2). While this probabilistic model does not have the correlation structure as in Mern et al. (and also Grosse et al. which we have discussed in the paper), our choice is inspired by the standard practice in Bayesian approximation of large models like Bayesian neural networks where very simple, independent priors such as $\mathcal{N}(0, \sigma^2 I)$ is employed. Note that this simplicity is necessary since having more complicated probabilistic models, e.g. via a GP or a Bayesian value network, are simply too expensive for decoding purposes. Note that we mentioned this as a limitation in Sec. 3.5, but we show good empirical performance and efficiency nevertheless.
> >
> > In any case, as suggested, we have rephrased “Bayesian optimization” in the text with “decision-making under uncertainty” whenever appropriate, thus weakening the link between ULTS and Bayesian optimization. Please let us know if this solves your concern.
> >
> >
> > > The other main concern is about the experiments (in the now Figure 5). The authors argue that ULTS lies at the Pareto frontier of all methods. However, nothing prevents ULTS to be tested with higher "average expanded nodes". Look at the Wikipedia dataset, I hardly believe that beam search finds the optimal decodings with 400 expanded nodes. So if ULTS is really useful, it should be able to find better decodings with 400 expanded nodes.
> >
> >
> > Please find new results with a larger $k_{max}$ for ULTS on Wikipedia with GPT2 and WMT-19 with T5 in Figure 14 and Figure 15 in Appendix C.10. We see that ULTS does indeed lie at the Pareto frontier in both cases. On the wikipedia dataset beam search eventually catches up.
> > We will add results with higher node expansions for other dataset-LLM combinations, too. Thanks for the suggestion.
> >
> > Importantly, the results in Figure 5. already encompass a realistic range of computational budgets, providing evidence that ULTS will be effective in practical applications. For instance, beam search with \(k = 5\) or \(k = 10\) represents a plausible choice (see  [1]) and one can already see that ULTS reaches this performance level.
> >
> > Please let us know if this solves your concern.
> >
> > **References**
> >
> > [1] Gian Wiher, Clara Meister, and Ryan Cotterell. On decoding strategies for neural text generators. TACL, 10, 2022

---

> > ### Author Response · Authors · 2024-11-25
> >
> > Thanks for your time and feedback so far. As the rebuttal phase is drawing to a close, we would appreciate any further comments you may have and would like to know if we have addressed your concerns. If so, we would be grateful if you could increase your score to reflect these changes.

---

> > > ### Comment · Reviewer_UmM7 · 2024-11-25
> > >
> > > Thank you for the additional experiments. It would be tremendous if all experiments include results with more average expanded nodes. It won't be negative if sometimes ULTS works worse than beam search, but those results can help the understanding of the proposed approaches.
> > >
> > > While I reserve my point about the Bayes part, I would not use it against this work now after weakening the link to BO. I have increased my scores accordingly.
> > >
> > > Since many parts have changed during the rebuttal, I suggest a thorough proofreading and adjustment in the next version of this work.

---

> > > > ### Author Response · Authors · 2024-11-25
> > > >
> > > > Thank you for adjusting your score and even more for your suggestions during the rebuttal. We will add results for larger k for the other datasets, too.

---

### Official Review · Reviewer_8FSd · 2024-11-03

**Soundness:** 4
**Presentation:** 3
**Contribution:** 4
**Rating:** 6
**Confidence:** 4

**Summary:**

This paper introduces Uncertainty-guided Likelihood-Tree Search (ULTS), a novel approach to decoding in large language models (LLMs) using Bayesian optimization. Unlike traditional myopic search methods (e.g., beam search), ULTS applies probabilistic reasoning, modeling uncertainty to prioritize search paths that maximize the likelihood efficiently. Experiments demonstrate that ULTS achieves comparable or better performance than baseline methods with fewer node expansions.

**Strengths:**

- The paper addresses a meaningful and interesting problem -- how to incorporate uncertainty in the search during sequential generation
- The method of viewing LLM decoding as bayesian optimization over trees is novel to me.
- The theoretical soundness is matched by well-executed experiments.
- The authors provide an open-source implementation, allowing easy adoption and further exploration.

**Weaknesses:**

- Symmetric priors may not fully capture natural language distributions although the authors have discussed this limitation.
- Although efficient in node expansions, ULTS has some overhead compared to batch-expanding methods like beam search.

**Questions:**

- How sensitive is ULTS to the choice of Dirichlet prior parameters?
- How can we further improve the runtime for ULTS?
- Is ULTS applicable to other tasks or customized to LLM decoding? For example, can we apply it to reinforcement learning tasks?

---

> ### Author Response · Authors · 2024-11-20
>
> Thank you for the review!
>
>
> > Symmetric priors may not fully capture natural language distributions although the authors have discussed this limitation.
>
> We agree. However, note that we also suggest an alternative choice for the prior, the "empirical" prior, which does not have the symmetric assumption of the Dirichlet prior since it’s sample-based. Nevertheless, even with the symmetric prior, we have already observed a good performance.
>
>
> > How sensitive is ULTS to the choice of Dirichlet prior parameters?
>
> We have an ablation for this in Figure 8 in the appendix. $\alpha$ indeed has a meaningful impact on the performance of ULTS: It can be seen as a hyperparameter that trades exploitation for exploitation (smaller value means more exploration). So, the choice of $\alpha$ should be based on how much budget one has since more exploration means more node expansions. If $\alpha$ is too small, ULTS might not use the full budget and, therefore does not achieve the performance it could have. But in terms of data efficiency (i.e. the sense of ULTS being closer to the Pareto front, trading-off number of node expansion with the likelihood value), we find ULTS to be beneficial for both small and large values of $\alpha$.
>
>
>
> > Although efficient in node expansions, ULTS has some overhead compared to batch-expanding methods like beam search. How can we further improve the runtime for ULTS?
>
> Beyond batching node expansions in the sense of batch BO, another idea might be to decode different prompts in parallel, i.e. run different tree searches and just batch nodes across different search trees (if the setting allows for this, e.g. if multiple users send requests at the same time). Moreover, in our newly added experiment on code completion with the HumanEval dataset, we show that ULTS can obtain better performance while being cheaper in terms of wall-clock than beam search. I.e., while ULTS expands nodes sequentially, it does so effectively.
>
> > Is ULTS applicable to other tasks or customized to LLM decoding? For example, can we apply it to reinforcement learning tasks?
>
> ULTS is customized to LLM decoding, but it's applicable to any kind of tree search problem (e.g. exploring a rollout tree in RL) as long as the reward of each node is a Categorical distribution (i.e., can be interpreted as a likelihood). We focus on LLMs since they naturally produce such a tree. In any case, we also provide a discussion on the application of ULTS in a more general, non-LLM Markov Decision Process via a toy example in Appendix C.1.

---

> ### Author Response · Authors · 2024-11-25
>
> Thanks for taking the time to review our paper. As the rebuttal phase is drawing to a close, we would appreciate any further comments you may have and would like to know if we have addressed your concerns.

---

> > ### Comment · Reviewer_8FSd · 2024-11-29
> >
> > Thank you for your response. I would like to keep my positive stance on this paper.

---

### Official Review · Reviewer_BdmK · 2024-11-04

**Soundness:** 2
**Presentation:** 3
**Contribution:** 2
**Rating:** 6
**Confidence:** 3

**Summary:**

This paper proposes Uncertainty-Guided Likelihood-Tree Search (ULTS), a novel probabilistic approach to decoding in large language models (LLMs). ULTS utilizes Bayesian optimization principles to guide search in a tree structure, using prior and posterior probability distributions to efficiently identify paths with high likelihood. By incorporating uncertainty into path selection, ULTS aims to balance computational efficiency with output quality, avoiding the need for exhaustive exploration. The approach is tested on machine translation, summarization, and text generation tasks, with results indicating improved log probability and BLEU scores compared to beam search and other decoding methods.

**Strengths:**

**Novel probabilistic approach**:
The proposed method, ULTS, introduces a Bayesian-inspired probabilistic tree search approach, efficiently incorporating uncertainty to improve path selection in language model decoding.

**Efficiency-focused design**:
The method reduces computational costs by leveraging prior and posterior distributions, balancing search depth with output quality.

**Weaknesses:**

**MAP Decoding Objective and Its Limitations in LLM Decoding**:
It has been widely observed that Maximum A Posteriori (MAP) decoding from language model generation, which this paper relies on, has notable limitations, such as its tendency to produce short, repetitive, or degenerate text [1,2,3]. While the authors acknowledge the issues and claim they are orthogonal to this paper in Section 3.5, decoding objectives in language models are crucially tied to LLM performance quality. These issues cannot be considered orthogonal as long as the main application is decoding from language models.

**Limited Discussion of Existing Decoding Methods**:
The Related Work section focuses on search algorithms for tree exploration but omits the discussion of standard decoding techniques such as top-k, nucleus, and MBR decodings, which are widely used in language model generation. The paper's contribution is difficult to comprehend without addressing these established methods and limitations of MAP decoding, particularly regarding generation quality and efficiency.

**Limited baselines and evaluation metrics**:
Strong baselines are missing while the experiments compare several recent decoding methods. For example, in close-ended generation tasks like NMT, state-of-the-art decoding methods such as Minimum Bayes Risk (MBR) are not evaluated. Comparing ULTS to MBR decoding would clarify its effectiveness in achieving high-quality translations. Summarization is mainly assessed through log probability, which may not sufficiently capture output quality regarding relevance, coherence, or informativeness. Including task-specific metrics like ROUGE, BLEURT, or human-evaluated coherence would provide a more comprehensive view of ULTS's performance in LLM applications.

[1] Felix Stahlberg, Bill Byrne; "On NMT Search Errors and Model Errors: Cat Got Your Tongue?," EMNLP-IJCNLP 2019.

[2] Bryan Eikema, Wilker Aziz; "Is MAP Decoding All You Need? The Inadequacy of the Mode in Neural Machine Translation,"  COLING 2020.

[3] Hugh Zhang, Daniel Duckworth, Daphne Ippolito, Arvind Neelakantan; "Trading Off Diversity and Quality in Natural Language Generation," HumEval 2021.

**Questions:**

See the weakness section above.
Additionally, although the above discusses issues with MAP decoding, relating this work to recent studies, such as the following, might make for an interesting contribution:

Davis Yoshida, Kartik Goyal, Kevin Gimpel; "MAP's not dead yet: Uncovering true language model modes by conditioning away degeneracy," ACL 2024.

---

> ### Author Response · Authors · 2024-11-20
>
> Thanks a lot for taking the time to review our paper! We uploaded a revised version. We would like to address your comments individually below:
>
> > **MAP Decoding Objective and Its Limitations in LLM Decoding:** It has been widely observed that Maximum A Posteriori (MAP) decoding from language model generation, which this paper relies on, has notable limitations, such as its tendency to produce short, repetitive, or degenerate text [1,2,3]. While the authors acknowledge the issues and claim they are orthogonal to this paper in Section 3.5, decoding objectives in language models are crucially tied to LLM performance quality. These issues cannot be considered orthogonal as long as the main application is decoding from language models.
>
>
> You are correct that likelihood maximization (MAP decoding) does not capture all the use cases in LLM decoding. We have moved our general formulation of ULTS’ acquisition function (expected utility function under posterior samples) in Sec. 5.4 to the main discussion in Sec. 3. Indeed, as we have shown in Sec. 5.4, ULTS’ probabilistic model can be paired with any utility function, such as one that includes repetition or degeneration penalty. Thank you for your suggestion!
>
> In any case, we also believe that vanilla likelihood maximization is also important. Since greedy/beam-search decoding is still the default in various applications such as code completion [1] and chemistry [2]; see also [3].
>
>
> > **Limited Discussion of Existing Decoding Methods:** The Related Work section focuses on search algorithms for tree exploration but omits the discussion of standard decoding techniques such as top-k, nucleus, and MBR decodings, which are widely used in language model generation. The paper's contribution is difficult to comprehend without addressing these established methods and limitations of MAP decoding, particularly regarding generation quality and efficiency.
>
> Thank you for the suggestion! We expanded the Related Work section.
>
>
> > Limited baselines and evaluation metrics: Strong baselines are missing while the experiments compare several recent decoding methods. For example, in close-ended generation tasks like NMT, state-of-the-art decoding methods such as Minimum Bayes Risk (MBR) are not evaluated. Comparing ULTS to MBR decoding would clarify its effectiveness in achieving high-quality translations. Summarization is mainly assessed through log probability, which may not sufficiently capture output quality regarding relevance, coherence, or informativeness. Including task-specific metrics like ROUGE, BLEURT, or human-evaluated coherence would provide a more comprehensive view of ULTS's performance in LLM applications.
>
> We added MBR as a new baseline. We also added new metrics, like ROUGE and Pass@1. The latter is for our newly added code-completion experiment on the HumanEval dataset. Please note that the six baselines we already have include recent methods like contrastive search (2022), speculative decoding (2023), and DoLA (2024).
>
>
> > Additionally, although the above discusses issues with MAP decoding, relating this work to recent studies, such as the following, might make for an interesting contribution:
> Davis Yoshida, Kartik Goyal, Kevin Gimpel; "MAP's not dead yet: Uncovering true language model modes by conditioning away degeneracy," ACL 2024.
>
> Thanks a lot for pointing this out! This work supports our view that plain likelihood maximization is still important, and ULTS is very capable of fulfilling this demand. We added this work to the paper.

---

> ### Author Response · Authors · 2024-11-25
>
> Thanks again for reviewing our paper.  As the rebuttal phase is drawing to a close, we kindly invite any additional comments you might have. We’d also like to confirm if we’ve successfully addressed your concerns. If so, we would greatly appreciate it if you could consider updating your score to reflect these improvements.

---

> > ### Comment · Reviewer_BdmK · 2024-11-25
> >
> > Sorry for the delayed response. Thank you for adding MBR as a baseline and including new metrics like ROUGE. These additions significantly enhance the evaluation. However, I noticed a few issues with the configuration of the MBR baseline:
> >
> > **Utility Function in MBR:**
> > In the original paper (Vamvas & Sennrich, 2024) for the MBR implementation used by the authors, COMET-22 (Rei et al., 2022) is recommended as the utility function for high-quality translation.
> >
> > **Number of Sampled Hypotheses:**
> > The original implementation in (Vamvas & Sennrich, 2024) uses 1024 sampled sentences to leverage MBR's potential fully. In your setup, the range of sampled sentences seems significantly smaller ({1, 2, 3, 4, 5, 10, 20}).
> >
> > **Sampling Strategy:**
> > The default epsilon sampling (epsilon_cutoff=0.02) is widely recognized as being better suited for MBR. The current use of temperature sampling (temperature=0.5) deviates from this standard and might affect the quality of the sampled hypotheses.
> >
> > **Evaluation Metrics:**
> > This point could be critical. While BLEU may be helpful, it primarily assesses surface-level similarity and cannot sufficiently capture the semantic accuracy and coherence of advanced decoding techniques like MBR and LLMs. I recommend incorporating semantic similarity metrics such as COMET-22 (Rei et al., 2022) or BLEURT (Sellam et al., 2020) to provide a more comprehensive evaluation of translation quality.
> >
> > - Rei et al., "COMET-22: Unbabel-IST 2022 submission for the metrics shared task," WMT 2022.
> > - Sellam et al., "BLEURT: Learning robust metrics for text generation," ACL 2020.

---

> ### Author Response · Authors · 2024-11-26
>
> Thanks so much for your reply! We are happy to hear that you find our additions significantly enhance the evaluation. We hope that your positive evaluation is reflected in your final score.
>
> Here, we will address your remaining comments.
>
> **MBR configuration**
> We followed exactly the setup of Bertsch et al., 2023 [2] (which is also one of the papers the MBR repository [3] mentions). The only things we changed were:
>
> * The metric, for fairness since we also evaluate on ROUGE. We therefore used fastChrF since this is the default in the repo and not used as evaluation metrics.
>
> * The number of samples, `1, 2, 3, 4, 5, 10, 20` instead of `30`. This is to be fair with other baselines that we used and roughly corresponds to the `beam_size` parameter. We would like to point out that our choice for the range of beam sizes is not particularly small. For instance, `k=5` or `k=10` represent plausible choices (see [1]). In fact, `num_samples=10` is the default choice in the repo.
>
> The aggregation method proposed by Vamvas & Sennrich, 2024 is currently not supported by the `mbr` package. It is planned to be released in v0.3.0 and the current version is v0.2.0, see [4]. Note that in the paper of Vamvas & Sennrich, 2024, they recommend the usage of the `mbr` package. While the reference code of Vamvas & Sennrich, 2024 is available, they are specifically for reproducing their paper and not for general usability. They stated this explicitly in their README [5].
> Hence, we followed Bertsch et al., 2023 instead since it seems to be the second latest paper to follow. Notice that they used temperature instead of `epsilon_cutoff`.
>
> In any case, we are happy to run further experiments with larger `num_samples`, `epsilon_cutoff = 0.02`, and use `COMET-22` as the metric. However it will take quite some time to gather the results, so we ask the reviewer for understanding. We will definitely add those results in our next revision.
>
> **Evaluation metrics**
>
> We added COMET and BLEURT for the machine translation experiment, see  Figure 14 and 15 in C.9. While ULTS performs well for small $k_{max}$, the performance decreases for larger $k_{max}$, unlike MBR. This might be due to the well-known miscalibration of the likelihood of LLMs as we already have discussed in our “Limitations” section. Note that in our other new experiment (HumanEval, Table 1), we show that ULTS performs positively even for small $k_{max}$ in terms of code-quality metric (Pass@1).
>
> In any case, ULTS’ acquisition function is general (see eq. 1 in our revised paper). Thus, ULTS can also be extended to use the utility function that takes into account of a reference set and a target metric akin to MBR. Moreover, ULTS’ can handle the modified likelihood as in [6] which seems to improve semantic similarity metrics. We added these directions as interesting future works in the revised version.
>
>
> **References**
>
> [1] Gian Wiher, Clara Meister, and Ryan Cotterell. On decoding strategies for neural text generators. TACL, 10, 2022
>
> [2] It's MBR All the Way Down: Modern Generation Techniques Through the Lens of Minimum Bayes Risk. Bertsch et al., 2023
>
> [3] <https://github.com/ZurichNLP/mbr/tree/main/experiments/bertsch-et-al-2023-mbr>
>
> [4] <https://github.com/ZurichNLP/mbr/tree/main?tab=readme-ov-file#changelog>
>
> [5] <https://github.com/ZurichNLP/mbr/tree/main/experiments/reference_aggregation>
>
> [6] Davis Yoshida, Kartik Goyal, Kevin Gimpel; "MAP's not dead yet: Uncovering true language model modes by conditioning away degeneracy," ACL 2024.

---

> > ### Comment · Reviewer_BdmK · 2024-11-29
> >
> > Thank you for your detailed response and the effort to address my concerns. I appreciate the significant enhancements made to the evaluation and have increased my score accordingly. Adding the planned experiments, ideally with larger generated sample sizes, would further strengthen the paper.
> >
> > I do not expect ULTS to outperform other decoding methods in every scenario due to LLM miscalibration, and it is fine if its performance decreases under certain conditions. What matters is providing a clear picture of ULTS's strengths and limitations under well-optimized settings.
> >
> > Additionally, I recommend prioritizing hyperparameter settings validated in the literature over repository defaults, as they often result in stronger benchmarks. Moreover, aligning the number of sampled hypotheses for MBR with the beam size for beam search seems less appropriate. Beam search typically degrades with larger `k`, whereas MBR benefits from increased sample sizes. Most studies treat these parameters independently, and I believe this approach would result in a fairer comparison. (The consistency between beam size and generated sample size in MBR might be beneficial in papers focusing on tight computational constraints, but this is not the case here.)

---

> > > ### Author Response · Authors · 2024-11-30
> > >
> > > Thanks again for your continued engagement and for updating the score!

---

### Official Review · Reviewer_bHVr · 2024-11-06

**Soundness:** 3
**Presentation:** 3
**Contribution:** 3
**Rating:** 6
**Confidence:** 4

**Summary:**

This paper introduces an efficient tree-search method by combining Bayesian Optimization with tree-search.

**Strengths:**

Authors propose an interesting idea of combining an existing sample efficient method: Bayesian Optimization (BO) for LLM generation.
BO is usually applied in continuous domain which use Gaussian priors. The authors extend BO and its acquisition function for LLM use-case. Search in LLM is an important problem.

The results based on log-likelihood show some proof of concept of the method being sample efficient.

**Weaknesses:**

experiments with regards to other known quantitative metrics apart from BLEU score and log-likelihood are lacking such as ROUGE-score

the explanation of the method can be further improved to better understand the contributions given there's no access to code.

tokens generated are much smaller which could be a limitation for tasks such as story-telling.

**Questions:**

the authors mention that high log-likelihood does not imply good quality or human desirable generation. In this regard Fig 3. showing BLEU scores is a nice result, however, it would be interesting to also observe ROUGE scores for summarization tasks.

For Fig. 3, is there a reason why BLEU score drops with increase in number of expanded nodes? Is this related to why ULTS works better for relatively short number of token generation?

From the plots in Fig. 4, it seems that by expanding more nodes, beam search or multinomial BS eventually give better log-likelihood, do the authors have any comment on this?

Also, have the authors thought about combining their method with beam search? for example: for initial few generations using BO based tree search and then afterwards continue on with beam-search like generation?

Can authors give a toy example on the how the N samples mentioned in Section 3.4 (decision making) is used by the acquisition function and correspondingly give some example value for the acquisition function ? (maybe if it is easier to add in Fig. 1 or if a separate Fig. is need? )

---

> ### Author Response · Authors · 2024-11-20
>
> **[1/2]**
>
> Thanks a lot for taking the time to review our paper! We would like to address your comments individually below:
>
> > experiments with regards to other known quantitative metrics apart from BLEU score and log-likelihood are lacking such as ROUGE-score …
> > … the authors mention that high log-likelihood does not imply good quality or human desirable generation. In this regard Fig 3. showing BLEU scores is a nice result, however, it would be interesting to also observe ROUGE scores for summarization tasks.
>
> Thank you for your suggestion. We added additional evaluations in the form of ROUGE for the existing experiments. Additionally, we added a further experiment on code generation (HumanEval dataset), where the Pass@1 metric is used to evaluate in addition to log-likelihood. Note that ULTS performs well in these cases.
>
> | Method                | Pass@1 | Log-lik  | Node expansion | Wall-clock   |
> | --------------------- | ------ | -------- | -------------- | ------------ |
> | Greedy                | 14,02% | \-28     | 399.369        | 13.557±0.493 |
> | Nucleus               | 14,63% | \-28,5   | 377.671        | 10.568±0.429 |
> | Contrastive           | 14,02% | \-28     | 399.366        | 9.981±0.334  |
> | Beamsearch-Mult (k=2) | 27,44% | \-24,375 | 496.963        | 7.235±0.420  |
> | ULTS (k_max=2)            | 25,00% | \-16,625 | 134,835        | 4.447±0.421  |
> | ULTS (k_max=3)            | 34,76% | \-15,5   | 146,866        | 4.936±0.454  |
> | ULTS (k_max=4)            | 32,32% | \-15,688 | 161,634        | 5.180±0.398  |
> | ULTS (k_max=5)            | 32,93% | \-14,938 | 179,39         | 5.596±0.358  |
> | ULTS (k_max=10)           | 31,71% | \-12,375 | 219,994        | 6.988±0.464  |
> | ULTS (k_max=20)           | 23,78% | \-12,062 | 180,36         | 6.041±0.369  |
>
>
>
> > Can authors give a toy example on the how the N samples mentioned in Section 3.4 (decision making) is used by the acquisition function and correspondingly give some example value for the acquisition function ? (maybe if it is easier to add in Fig. 1 or if a separate Fig. is need? )
>
> Yes. We added a new figure (Figure 2) to the method section; please refer to the updated text. It shows the first two iterations of results in a toy example of a tree of depth and width equal to 3. We show the observation (the categorical distribution), the sampled approximation of the posterior, and the acquisition function derived from the posterior. We hope that this clarifies any potential confusion.
>
> > the explanation of the method can be further improved to better understand the contributions given there's no access to code.
>
> The code is available as part of the supplementary material. It includes a README and examples on how to run it in combination with the HuggingFace library. We hope this, in combination with Fig. 2, is sufficient for a better understanding of our method.
>
> We do, of course, aim for the presentation to be understandable without the need to access the code, though. Beyond the figure you thought would be helpful, we also edited parts of the paper based on the feedback from the other reviewers. However, if there are any other sections that are less clear, we are happy to make further changes.
>
>
> > tokens generated are much smaller which could be a limitation for tasks such as story-telling.
>
> We added a comparison with a larger number of max generated tokens (500 on the new HumanEval experiment). The results are comparable to other experiments.
>
>
> > For Fig. 3, is there a reason why BLEU score drops with increase in number of expanded nodes? Is this related to why ULTS works better for relatively short number of token generation?
>
> This is something that has also been observed for beam search in related work, see [1, 2]. So we believe, it is due to the likelihood of not perfectly correlating with other metrics, like the BLEU score. Nevertheless, we have shown that ULTS performs well across various non-likelihood metrics.

---

> ### Author Response · Authors · 2024-11-20
>
> **[2/2]**
>
> > From the plots in Fig. 4, it seems that by expanding more nodes, beam search or multinomial BS eventually give better log-likelihood, do the authors have any comment on this?
>
> Yes, more node expansions imply better log-likelihood since this implies more exploration, i.e. closer to exhaustive search. However, this is undesirable in practice since cost is also a critical consideration when choosing a decoding method. To obtain a more holistic view of this, we need to consider trade-offs between performance and cost, which is why we showed the plots in our Experiments section: ULTS is close or in the **Pareto frontier** in each of the plot, indicating that ULTS achieves the best performance-cost trade-off compared to the baseline methods.
>
> > Also, have the authors thought about combining their method with beam search? for example: for initial few generations using BO based tree search and then afterwards continue on with beam-search like generation?
>
> This is an interesting idea! The combination of a smarter tree search method and beam search has been studied in the context of Monte Carlo Tree Search, where beam search is used for rollout [3]. In future work, ULTS can be extended by including rollouts (e.g. when rewards that are only observable at the leaves are available). In this case, beam search can combined with ULTS.
>
> **References**
>
> [1] Gian Wiher, Clara Meister, and Ryan Cotterell. On decoding strategies for neural text generators.
> TACL, 10, 2022.
>
> [2] Yang, Yilin, Liang Huang, and Mingbo Ma. "Breaking the beam search curse: A study of (re-) scoring methods and stopping criteria for neural machine translation." EMNLP 2018.
>
> [3] Zhang, Shun, et al. "Planning with large language models for code generation." ICLR 2023.

---

> ### Author Response · Authors · 2024-11-25
>
> Thank you for your feedback and the time you’ve dedicated so far. As the rebuttal phase comes to an end, we welcome any further comments you may have. If we’ve successfully addressed your concerns, we would be grateful if you could consider adjusting your score to reflect the changes.

---

> ### Author Response · Authors · 2024-11-30
>
> Just a quick reminder that as a reviewer, you can only post replies until December 2nd. If you have any remaining questions, please don't hesitate to let us know.

---

> > ### Comment · Reviewer_bHVr · 2024-12-02
> > **Response to Authors**
> >
> > Hi,
> >
> > Thanks a lot for your response, I've updated my ratings

---

### Author Response · Authors · 2024-12-03
**To all reviewers**

As the discussion period comes to an end and there appear to be no remaining open questions, we would like to **thank you for the productive discussion** and take this opportunity to summarize the main changes incorporated into the revised version(s). We appreciate that you acknowledged these changes by now rating the paper unanimously on the side of acceptance, either by increasing your score or keeping your positive one:

* We added evaluations using additional metrics, including ROUGE scores, COMET, BLEURT, and Pass@1.

* We conducted a new experiment for a code completion task.

* We incorporated MBR as a baseline and extended the related work section.

* We included evaluations for larger $k_{max}$ values and will continue this for the remaining datasets.

* We improved the presentation by adding Figure 2, restructuring parts of the method section, de-emphasizing the relation to Bayesian optimization, and making the backup step more explicit by including the corresponding formulas.

* We added a section detailing all hyperparameters of the baselines, rather than referring to them simply as HuggingFace’s defaults.

---

### Meta-Review · Area_Chair_LgxZ · 2024-12-23

**Metareview:**

This work proposes a value-based decoding algorithm, called Uncertainty-guided Likelihood-Tree Search (ULTS). The decoding is done in a two-step manner. The first step is Algorithm 1 in Appendix D where a prior belief over the value is learned, and then it is used as a cheap heuristic at decoding time to guide the search. This algorithm is used specifically for performing better maximum-likelihood decoding compared to beam search and it is shown to produce sequences with higher likelihood compared to beam search with similar run-time efficiency. Overall, all reviewers found the paper to be interesting and the authors' rebuttal addressed the concerns of the reviewers. In AC's own reading of the paper and the subsequent discussion with the reviewers, we realized that the paper effectively introduces a new controlled decoding algorithm, see (Yang and Klein, 2021), (Deng and Raffel, 2023), (Mudgal et al., 2023), and (Han et al., 2024). In all of these works, a value-function or q-function is learnt offline that is then used for guiding decoding. If the q-function is learnt with a reward that is log-likelihood, then controlled decoding is effectively solving maximum likelihood decoding. This connection leaves two important questions: (1) For the pure case of maximum likelihood decoding, controlled decoding should be used as a competing baseline; (2) Given that most modern applications of generative language models use a variant of temperature sampling, the impact of this work would be more pronounced if the authors applied their method to more general rewards (like helpfulness, harmlessness, etc). The authors themselves mention that there is nothing that makes the proposal specific to maximum likelihood decoding and this extension will further strengthen the work and help situate it in the literature.

Overall, the authors' proposal has two major pieces: *value function learning* (which is done for likelihood) and *inference-time search.* Both of these methods need to be substantiated separately as solid algorithms in an ablation study for the contribution of the paper to be sufficiently substantiated. The aforementioned controlled decoding methods all learn value functions and the value function learning proposal in this paper needs to be substantiated against those methods. For a given value function, the search method proposed herein also needs to be ablated against other search methods, like A* search, blockwise best-of-k, etc. As such, given that these critical ablations are missing, the paper is recommended to be rejected so that the authors can substantiate these points effectively. We understand that this decision may be disappointing to the authors given the consensus positive scores of the reviewers, however, we would like to clarify that after extensive discussion between the SAC, the AC, and the reviewers, we all agreed with this decision.

Yang, Kevin, and Dan Klein. "FUDGE: Controlled text generation with future discriminators." arXiv preprint arXiv:2104.05218 (2021).

Deng, Haikang, and Colin Raffel. "Reward-augmented decoding: Efficient controlled text generation with a unidirectional reward model." arXiv preprint arXiv:2310.09520 (2023).

Mudgal, S., Lee, J., Ganapathy, H., Li, Y., Wang, T., Huang, Y., Chen, Z., Cheng, H.T., Collins, M., Strohman, T. and Chen, J., 2023. Controlled decoding from language models. arXiv preprint arXiv:2310.17022.

Han, S., Shenfeld, I., Srivastava, A., Kim, Y. and Agrawal, P., 2024. Value Augmented Sampling for Language Model Alignment and Personalization. arXiv preprint arXiv:2405.06639.

**Additional Comments On Reviewer Discussion:**

The paper has received positive consensus from the reviewers but during the AC-reviewer discussion it became clear that an important connection between this work and controlled decoding was missing, and also the paper still needed key ablations before it would become publishable. Due to this issue, in close consultation with the SAC, the AC has decided to recommend rejection of the paper and the four reviewers who participated in the AC-reviewer discussion agreed with the decision.

---

### Decision · Program_Chairs · 2025-01-22

Reject